# REMOTELY DETECTABLE ROBOT POLICY WATERMARKING

**Michael Amir,**[*] **Manon Flageat**[*] **& Amanda Prorok**
University of Cambridge
`{ma2151,mf873,asp45}@cam.ac.uk`

## ABSTRACT

The success of machine learning for real-world robotic systems has created a new form of intellectual property: the trained policy. This raises a critical need for novel methods that verify ownership and detect unauthorized, possibly unsafe misuse. While watermarking is established in other domains, physical policies present a unique challenge: remote detection. Existing methods assume access to the robot's internal state, but auditors are often limited to external observations (e.g., video footage). This "Physical Observation Gap" means the watermark must be detected from signals that are noisy, asynchronous, and filtered by unknown system dynamics. We formalize this challenge using the concept of a *glimpse sequence*, and introduce Colored Noise Coherency (`CoNoCo`), the first watermarking strategy designed for remote detection. `CoNoCo` embeds a spectral signal into the robot's motions by leveraging the policy's inherent stochasticity. To show it does not degrade performance, we prove `CoNoCo` preserves the marginal action distribution. Our experiments demonstrate strong, robust detection across various remote modalities—including motion capture and side-way/top-down video footage—in both simulated and real-world robot experiments. This work provides a necessary step toward protecting intellectual property in robotics, offering the first method for validating the provenance of physical policies non-invasively, using purely remote observations.

## 1 INTRODUCTION

The rise of machine learning in robotics has yielded high-performance policies capable of sophisticated locomotion, manipulation, and navigation (Lee et al., 2020; Smith et al., 2023; Hoeller et al., 2024). These policies, often deep neural networks resulting from significant investment, represent a critical new form of intellectual property (IP). As commercial deployment accelerates, the risk of unauthorized misuse and theft escalates, creating an urgent need for reliable methods to verify ownership and provenance.

Digital watermarking is the standard mechanism for IP protection in domains like multimedia (Cox et al., 1997) or large language models (Kirchenbauer et al., 2023; Dathathri et al., 2024). However, existing methods for watermarking policies (Behzadan & Hsu, 2019; Chen et al., 2021) suffer from a critical limitation: they assume *white-box* access to the system, requiring direct inspection of internal states, action logs, or specific trigger environments. In realistic scenarios (Figure 1), such access is often impossible or untrustworthy.

The ability to verify policy provenance using only remote observations (e.g., CCTV footage) is essential not only for IP protection but also for high-impact AI safety applications. For instance, remote detection enables *Scalable Safety Compliance*, allowing regulators to non-invasively verify whether safety-critical systems (e.g., autonomous vehicles) are authorized for deployment by checking them against a database of certified policy signatures. It also facilitates *Trustworthy Forensics and Accountability*. Following an incident (e.g., an autonomous vehicle crash or industrial accident), determining the provenance of the deployed control software is critical for liability. Relying solely

---

[*]Equal contribution in alphabetical order.
Project website and code: `https://sites.google.com/view/robotpolicywatermarking/`

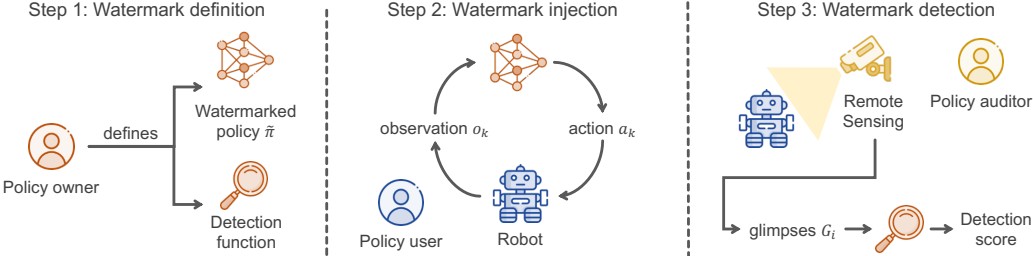

Figure 1: Overview of the pipeline for robot policy watermarking. In Step 1, the *policy owner* trains a policy, adds a watermark to it and produces a detection function to identify it. In Step 2, the watermarked policy is used by a *policy user* who deploys it on their own robot. In Step 3, a *policy auditor* aims to identify the policy used on the robot. To do so, they can only access glimpses of the policy behaviour through remote sensing, such as a camera feed; these glimpses are passed through the detection function to identify the policy.

on onboard logs is insufficient, as they may be unavailable due to damage or deliberately tampered with by adversarial actors seeking to evade responsibility or fraudulently claim damages. Remote watermark detection provides an independent mechanism for auditors to verify the active policy using external, tamper-resistant data sources like traffic camera footage.

These scenarios introduce a fundamental challenge we term the *Physical Observation Gap*. The auditor does not observe the policy's actions (say, torque commands), but rather their consequences (say, movement captured by camera). The watermark signal must cross this gap, surviving severe distortions that destroy traditional watermarking signatures. This entails three primary challenges: (C1) *Synchronization Uncertainty* between the policy's clock and the remote sensor; (C2) *System Dynamics* filtering the actions through the robot's complex, unknown physics (e.g., inertia, friction); and (C3) *Interference and Noise* from the robot's primary behavior and the environment.

In this work, we introduce **Co**lored **No**ise **Co**herency (`CoNoCo`), the first watermarking strategy for robot policy designed to enable remote watermark detection. `CoNoCo` operates in the frequency domain. It embeds the watermark by exploiting the inherent stochasticity of standard continuous control policies. The watermark is then detected using *Spectral Coherency*, a normalized frequency-domain metric conceptually analogous to a correlation coefficient for specific frequencies. It possesses a notable invariance property that cancels out the filtering effects of unknown system dynamics (Theorem 5.2). This allows us to, e.g., inject the watermark on the level of torque commands executed by the robot, but detect it even when we only observe noisy, video footage-derived velocity estimates. By combining this invariance with explicit synchronization techniques, `CoNoCo` achieves robust detection despite the challenges of the Physical Observation Gap.

Our contributions are summarized as follows. **(i)** We formalize the problem of remotely detectable policy watermarking using the concept of *glimpse sequences* and characterize the fundamental challenges posed by the Physical Observation Gap (C1-C3). **(ii)** We introduce `CoNoCo`, a novel frequency-domain strategy based on colored noise injection and spectral coherency detection. To our knowledge, it is the first method capable of verifying policy provenance using only remote measurements. **(iii)** Finally, we demonstrate `CoNoCo`'s effectiveness in simulated and real-world robot tasks with challenging detection modalities like motion capture, and top-down, and sideway video footage. We compare it to adapted variants of existing watermarks, and show its robustness to different types of noise and interference (including deliberate adversarial noise).

## 2 RELATED WORK

**Digital Watermarking and Signal Processing.** Watermarking has historically been used for IP protection in multimedia (Berghel & O'Gorman, 1996; Swanson et al., 1998), with methods such as Spread Spectrum (Cox et al., 1997) embedding robust, imperceptible signals across wide frequency bands. More recently, watermarking has been applied to generative models (Wen et al., 2023; Kirchenbauer et al., 2023; Dathathri et al., 2024). These methods assume access to well-behaved

signals and struggle with the dynamics of remote physical systems; in this work, we extend these frequency-domain principles to such challenging settings.

**Watermarking in Cyber-Physical Systems (CPS).** In CPS security, "dynamic watermarking" defends robots against sensor attacks by superimposing signals onto control inputs (Satchidanandan & Kumar, 2016; Ko et al., 2016). While related, these methods target real-time security (integrity) rather than IP (provenance) and crucially assume auditor access to internal control signals.

**Watermarking Neural Networks (NN).** Methods exist to verify ownership of NN models by embedding signatures into weights (Darvish Rouhani et al., 2019) or using rare "trigger" inputs (backdooring) (Adi et al., 2018; Szyller et al., 2021). These require white-box access or the ability to actively query the model.

**Watermarking Policies and Agents.** Prior work on policy watermarking typically modifies behavior in specific situations. Behzadan & Hsu (2019) requires execution in a secret "trigger environment.", Chen et al. (2021) enforces secret actions in specific "safe states." Methods for agentic systems (Huang et al., 2025) watermark high-level behavior. Unlike `CoNoCo`, all of these approaches require access to the internal state of the policy, making them unsuitable for remote detection.

## 3 PROBLEM STATEMENT

We address the challenge of watermarking a stochastic robotic control policy $\pi_\theta$ so that the watermark can be detected using only remote measurements. The policy maps observations $\mathbf{o}_k \in \mathcal{O}$ to actions $a_k \in \mathcal{A}$ at discrete time steps $k = 0, 1, \ldots$. We assume a standard structure common in continuous control Reinforcement Learning (RL), where the policy outputs the parameters of a Gaussian distribution[1]: $a_k = \mu_\theta(\mathbf{o}_k) + \Sigma_\theta(\mathbf{o}_k)\epsilon_k$. Here, $\mu_\theta$ is the mean action (the primary behavior), and $\Sigma_\theta(\mathbf{o}_k)$ determines how much the action deviates from the mean, which we call the *exploration scale*. $\epsilon_k \sim \mathcal{N}(0, I)$ is White Gaussian Noise (WGN)—random, uncorrelated noise used for exploration. A robot, $\mathcal{R}$, executes this policy.

The objective is to create a watermarked policy $\widetilde{\pi}_\theta$ using a secret key $\mathcal{K}$. An *auditor*, possessing $\mathcal{K}$, must detect this signature using only remotely collected, passive data (e.g., video footage), without access to the robot's internal state. This is difficult due to the challenges (C1-C3) and requirements (W1-W2) below.

Firstly, we must overcome the "Physical Observation Gap": the separation between the digital policy execution and the remote physical observations. This gap introduces three primary challenges:

C1. **Synchronization Uncertainty.** The policy executes at an internal rate $f_\pi$, which is often unknown (within bounds $[f_{\pi,\mathrm{lb}}, f_{\pi,\mathrm{ub}}]$) and may vary (jitter). Remote sensors (rate $f_g$) sample independently. This causes frequency misalignment. Furthermore, the recording might start at an arbitrary time after the policy execution begins, introducing an unknown time offset.

C2. **System Dynamics.** The auditor does not see the policy commands (e.g., motor torques); they see the physical response (e.g., movement), which is transformed by the robot's unknown and time-varying physical dynamics $\mathcal{S}_{\mathrm{dyn}}$. The physics filter and distort the original signal.

C3. **Interference and Noise.** The policy behaviour $\mu_k$ is typically much stronger than the watermark signal, acting as significant interference. External disturbances and sensor noise further corrupt the observation.

We model the Physical Observation Gap through a **Glimpse Sequence Formalism**. The policy runs at unknown times $\{T_k\}$. The executed action $a_{\mathrm{exec}}(t)$ drives the robot's state evolution $\dot{s}(t) = \mathcal{S}_{\mathrm{dyn}}(s(t), a_{\mathrm{exec}}(t))$.

**Definition 3.1** (Glimpse Sequence). *A remote sensor samples the state at times $\{t_i\}$ (rate $f_g$), producing measurements $G_i = \mathcal{G}_{\mathrm{map}}(s(t_i)) + \eta_i$, where $\eta_i$ is measurement noise. The sequence $G = (G_i)_{i=0}^{N-1}$ is the **glimpse sequence**, the sole data available for detection. $\mathcal{G}_{\mathrm{map}}$ is a function that*

---

[1]Often, the action is bounded by a saturation function (e.g., $a_k = \tanh(\ldots)$). For simplicity, we omit this in our theoretical analysis; however, our experiments show that detection remains effective even when such a function is applied.

*maps the state of the system at time $t_i$ to a remote observation, such as a velocity estimate from video data. In complex (MIMO) systems, $G$ is multi-dimensional; we denote the $d$-th dimension as $G_d$.*

Secondly, to be useful, the watermarked policy $\widetilde{\pi}_\theta$ must satisfy two requirements:

W1. **Marginal Distribution Preservation.** To ensure the watermarked policy behaves like the original, we require the probability distribution of actions to remain unchanged: $p_{\pi_\theta}(a|\mathbf{o}) = p_{\widetilde{\pi}_\theta}(a|\mathbf{o})$ for all $\mathbf{o}, a$.

W2. **Robust Detectability.** The detector $D_{\mathcal{K}}(G)$ must reliably distinguish $\widetilde{\pi}_\theta$ from $\pi_\theta$ despite C1-C3.

## 4    THE COLORED NOISE COHERENCY STRATEGY (CoNoCo)

The distortions caused by the Physical Observation Gap (C1-C3) make detection methods based on precise timing (time-domain) fragile. We introduce Colored Noise Coherency (CoNoCo), a strategy that analyzes the signal's frequency content (frequency-domain), which is more robust to these distortions. CoNoCo operates on two principles *(i)* It embeds the watermark by replacing the WGN exploration noise with normalized Colored Gaussian Noise (CGN). CGN is a type of "shaped" noise that concentrates energy in a target frequency band. We show in Section 5 that this approach improve detection while satisfying W1. *(ii)* It detects this signature using Spectral Coherency, a technique robust to unknown dynamics, combined with synchronization methods to address remote sensing challenges.

**Watermark Generation and Injection.**    We generate the watermark by creating a CGN sequence, $W_k$, to replace the original WGN $\epsilon_k$. The watermark is defined by a secret key $\mathcal{K} = \{S, \mathcal{B}\}$, where $S$ is a secret seed and $\mathcal{B} = [f_{\min}, f_{\max}]$ is a frequency band (e.g., in Hz). The process is described in Algorithm 1.

To generate $W_k$, we filter a pseudorandom WGN sequence $X$ (derived from $S$) using a digital Band-Pass filter H. The goal is for the physical actions to vibrate within $\mathcal{B}$. Since the physical frequency depends on the uncertain policy rate $f_\pi$ (C1), the digital filter spans $[f_{min}/f_{\pi,ub}, f_{max}/f_{\pi,lb}]$. This guarantees the resulting physical signal covers $\mathcal{B}$, regardless of $f_\pi$. $X$ is then filtered and normalized to produce $W_k$.

The watermarked policy $\widetilde{\pi}_\theta$ utilizes this CGN: $\tilde{a}_k = \mu_\theta(\mathbf{o}_k) + \Sigma_k \cdot W_k$. The advantage of targetting a specific band $\mathcal{B}$ is that it can be chosen outside the anticipated frequencies spectrum of $\mu_\theta(\mathbf{o}_k)$, reducing policy interference (C3). Note that the time-varying exploration scale $\Sigma_k$ changes the amplitude of the watermark; we analyze this effect in Section 5. If the policy action is multi-dimensional, we generate independent CGN sequences for each dimension to improve detection.

**Watermark Detection Strategy.**    The detection strategy (Algorithm 1) aims to address the Physical Observation Gap. We first address the challenge of an unknown policy frequency (C1). The unknown policy frequency $f_\pi$ must be found. The detector searches over a grid of candidate frequencies $\mathcal{F}_{\text{search}} \subseteq [f_{\pi,\text{lb}}, f_{\pi,\text{ub}}]$. For each candidate $s$, the detector regenerates the watermark $W$ and resamples (time-stretches) it from the hypothesized rate $s$ to the known glimpse rate $f_g$, yielding a hypothesis $W'_s$. This aligns the time scales. We next address (C2). Detecting the watermark after it passes through unknown system dynamics is challenging. We use Spectral Coherency, a frequency-domain metric analogous to correlation in statistics.

**Definition 4.1** (Complex Coherency). *The Complex Coherency $C_{XY}(f)$ between two processes X and Y at frequency $f$ is defined as: $C_{XY}(f) = \frac{S_{XY}(f)}{\sqrt{S_{XX}(f)S_{YY}(f)}}$. Here, $S_{XX}(f)$ and $S_{YY}(f)$ are the* Power Spectral Densities (PSDs)*, representing the energy (variance) of X and Y at frequency $f$. $S_{XY}(f)$ is the* Cross-Spectral Density (CSD)*, representing the covariance between X and Y at frequency $f$.*

Coherency is a normalized version of the CSD. Its magnitude $|C_{XY}(f)| \in [0, 1]$ acts like a correlation coefficient at frequency $f$, indicating the strength of the linear relationship between $X$ and $Y$. Crucially, if $Y$ is the output of a Linear Time-Invariant (LTI) system $H$ with input X, then $|C_{XY}(f)| = 1$, regardless of the specifics of $H$ (Theorem 5.2). This property makes CoNoCo robust to system dynamics. For example, if the glimpses are related to the robot's actions by an ODE with

constant coefficients (e.g., robot executes torque commands but we observe velocity glimpses), this transformation is LTI, and does not impact coherency.

For a given hypothesis $s$, we calculate the coherency between $W'_s$ and $G$. Let $C_{W'_d G_d}(f; s)$ be the coherency between the $d$-th dimension of $W'_s$ and the glimpse $G_d$. The final detection score is the maximum average magnitude within the secret band $\mathcal{B}$, calculated using Welch's method (Karl, 2012), which breaks the signal into segments, and optimized over all hypotheses: $D(G) = \max_{s \in \mathcal{F}_{\text{search}}} \left( \frac{1}{D} \sum_{d=1}^{D} \text{mean}_{f \in \mathcal{B}} |C_{W'_d G_d}(f; s)| \right)$.

---

**Algorithm 1** Watermark generation and detection procedures.

| **Watermark Generation** | **Watermark Detection** |
|---|---|
| **Require:** Seed $S$, Length $N$, Dimensions $D$, Band $\mathcal{B} = [f_{min}, f_{max}]$, Freq Bounds $[f_{\pi,\text{lb}}, f_{\pi,\text{ub}}]$ | **Require:** Glimpses $G$, Glimpse Freq $f_g$, Key $\mathcal{K} = \{S, \mathcal{B}\}$, Search Range $\mathcal{F}_{\text{search}}$ |
| 1: $W \leftarrow \text{Zeros}(N, D)$ | 1: $D \leftarrow \text{NumDimensions}(G)$; $BestScore \leftarrow 0$ |
| 2: $B_{low} \leftarrow f_{min}/f_{\pi,\text{ub}}$; $B_{high} \leftarrow f_{max}/f_{\pi,\text{lb}}$ | 2: $W_{base} \leftarrow \text{RegenerateWatermarkSequence}(\dots)$ |
| 3: $H_{\mathcal{B}} \leftarrow \text{DesignButterworthBPF}(B_{low}, B_{high})$ | 3: **for** $s \in \mathcal{F}_{\text{search}}$ **do** $\quad\triangleright$ Frequency Alignment |
| 4: **for** $d = 1$ to $D$ **do** | 4: $\quad W'_s \leftarrow \text{ResamplePoly}(W_{base}, f_g/s)$ |
| 5: $\quad S_d \leftarrow \text{DeriveDimSeed}(S, d)$ | 5: $\quad Score_{sum} \leftarrow 0$ |
| 6: $\quad X \leftarrow \text{GenerateWGN}(S_d, N)$ | 6: $\quad$ **for** $d = 1$ to $D$ **do** |
| 7: $\quad W_{\text{raw}} \leftarrow \text{ApplyLTIFilter}(H_{\mathcal{B}}, X)$ | 7: $\quad\quad W'_d \leftarrow W'_s[0 : |G|, d]$ |
| 8: $\quad W[:, d] \leftarrow W_{\text{raw}}/\text{Std}(W_{\text{raw}})$ $\quad\triangleright$ Normalize | 8: $\quad\quad f, C_d \leftarrow \text{Coherency}(G[:, d], W'_d, f_g)$ |
| 9: **return** $W$ | 9: $\quad\quad Score_d \leftarrow \text{Mean}(\text{Abs}(C_d[f \in \mathcal{B}]))$ |
| | 10: $\quad\quad Score_{sum} \leftarrow Score_{sum} + Score_d$ |
| | 11: $\quad$ **if** $Score_{sum}/D > BestScore$ **then** |
| | 12: $\quad\quad BestScore \leftarrow Score_{sum}/D$ |
| | 13: **return** $BestScore$ |

---

Returning to (C1), we must also consider the possibility of a time offset, where the glimpse sequence starts being recorded some time after the policy has already begun being executed, hence the glimpse data does not collect some initial fraction of the $W_k$ sequence. While Algorithm 1, as described here, assumes the glimpse sequence begins at minimal time offsets from the start of the robot's policy execution, we can robustly address time offsets using the Generalized Cross-Correlation Phase Transform (GCC-PHAT). This is shown in Appendix G.1.

## 5 Analysis of Watermark Properties

We now study the properties of `CoNoCo`. Theorem proofs are provided in the Appendix.

**(W1) Marginal Distribution Preservation and Utility.** We show that the watermark injection process preserves the statistical distribution of the exploration noise (Theorem 5.1), proving requirement (W1).

**Theorem 5.1.** *Let $W_k$ be generated by filtering a WGN sequence $X_k \sim \mathcal{N}(0, I)$ through a stable LTI filter $H$, followed by normalization to unit variance. Then the marginal distribution of $W_k$ is also $\mathcal{N}(0, I)$.*

Theorem 5.1 establishes that $p_{\pi_\theta}(a|\mathbf{o}) = p_{\tilde{\pi}_\theta}(a|\mathbf{o})$; the statistics of the actions at any single time step are identical for the watermarked and original policies. However, using CGN instead of WGN introduces *temporal autocorrelation*: the noise at one time step is related to the noise at previous steps, rather than being fully independent. We empirically show that this does not affect system performance if the band $\mathcal{B}$ is chosen appropriately. The reason CGN is often benign is noted in (Lillicrap et al., 2015): temporally correlated noise can lead to smoother exploration, often improving performance in continuous control.

**(W2) Robust Detectability.** Detectability relies on the properties of Spectral Coherency. We first start with idealized assumptions, assuming constant dynamics (LTI) and constant exploration scale ($\Sigma_k = \Sigma$). The system mapping the watermark $W_k$ to the glimpse $G_i$ is LTI (Linear Time-Invariant). The core reason we use coherency for detection is that coherency can "see through" the dynamics; e.g., even if the robot uses torque actions but we observe velocity glimpses, the detection score is not affected by this transformation. This is expressed in the following, well-known theorem (Karl, 2012):

**Theorem 5.2** (Invariance of Coherency Magnitude under LTI Filtering). *Let X and Y be stationary processes related by an LTI system $H_{sys}$. In the absence of noise, $|C_{XY}(f)| = 1$, regardless of $H_{sys}$ (provided $H_{sys}(f) \neq 0, S_{XX}(f) > 0$).*

In practice, the glimpse $G$ is corrupted by interference from $\mu_k$ and sensor noise $\eta_i$ (C3). We quantify the effect of this interference using the Signal-to-Interference-plus-Noise Ratio (SINR).

**Definition 5.1** (Signal-to-Interference-plus-Noise Ratio (SINR)). *The **SINR** at frequency $f$ is the ratio of the desired signal power to the power of the undesired components: $SINR(f) = \frac{P_S(f)}{P_N(f)}$. Here, $P_S(f)$ is the power (PSD) of the watermark signal in the glimpse, and $P_N(f)$ is the power of the interference from the policy $\mu_k$, sensor noise, and other sources.*

**Theorem 5.3** (SINR in Watermarked Policies). *Consider the watermarked policy action $\tilde{a}_k = \mu_k + \Sigma W_k$, with constant $\Sigma$, driving an LTI system $H_{sys}$. Assume $W$, $\mu$, and measurement noise $\eta$ are mutually independent. Then the magnitude squared coherency between $W$ and the glimpse $G$ is $|C_{WG}(f)|^2 = \frac{SINR(f)}{SINR(f)+1}$.*

Theorem 5.3 links detectability and SINR. Coherency approaches 1 when the watermark power $P_S(f)$ is significantly greater than the noise power $P_N(f)$. The exploration scale $\Sigma$ directly controls the strength of $P_S(f)$; thus, policies with more exploration (larger $\Sigma$) allow for better detectability (W2)[2].

The above analysis holds in idealized (LTI) conditions. Real robotic systems are often LTV (Linear Time-Varying), with changing dynamics $\mathcal{S}_{dyn}(t)$ and scale $\Sigma_k$. A time-varying $\Sigma_k$ causes "spectral smearing," reducing SINR. Furthermore, we use Short-Time analysis (Karl, 2012), assuming slow-changing system dynamics. Rapid changes, particularly in phase, can bias the coherency estimate downwards. CoNoCo mitigates this via its aggregation strategy (Section 3): (i) averaging over multiple glimpse dimensions can pick up signal from dimensions that behave more like an LTI; (ii) the band $\mathcal{B}$ can be chosen to avoid unstable frequencies. Please see Appendix C for details and practical tuning tips in Appendix D.

## 6 EXPERIMENTAL SETUP

To evaluate CoNoCo, we design experiments covering multiple glimpse modalities and environments (Figure 2). We categorize glimpse modalities based on the auditor's access (Table 1). **Ground Truth Action** assumes direct access to the watermarked action signal. This unrealistic setting serves as an idealized baseline, free from the challenges (C1-C3) central to this work. **Onboard Sensors** use proprioceptive measurements from the robot. This modality is affected by system dynamics and noise (C2, C3), as it only estimates the effect of the actions after these are filtered by physics. *Remote* modalities use fully external sensors and are the main focus of this work, encompassing all challenges (C1-C3). **Remote Motion Capture** *remotely* approximates the motion of the robot using multiple cameras; its readings are unsynchronised with the policy (C1). **Remote Camera Feed** uses a single-pov video recording at either top-down or sideways angle; it is similar to Motion Capture but typically provides less precise estimates (higher C3). Across our onboard sensor, motion capture, and remote camera settings, $f_g/f_\pi \approx 5$. Other ratios attained similar performance.

Table 1: Overview of the glimpse modalities considered in our experiments and the challenges they induce.

| Challenge | Ground Truth Action | Onboard Sensors | Motion Capture | Camera Feed |
|---|---|---|---|---|
| C1: Sync. Uncertainty | - | - | ⚠ | ⚠ |
| C2: System Dynamics | - | ⚠ | ⚠ | ⚠ |
| C3: Interference & Noise | - | ⚠ | ⚠ | ⚠ |

As CoNoCo is the first strategy for remote detection, no direct baselines exist. To be able to perform comparisons, we therefore introduce three adapted variants of watermarks intended for related settings:

---

[2]We emphasize that Theorems 5.2 and 5.3 are well-known properties of LTI systems (Karl, 2012). The contribution of CoNoCo is in exploiting these properties to design a remotely detectable watermarking strategy

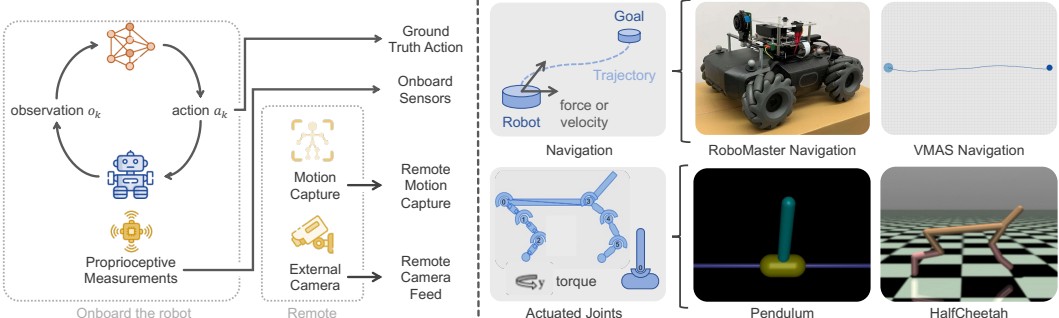

Figure 2: Overview of the Experimental Setup. (Left) Glimpse modalities: *Ground Truth Action* uses the watermarked action signal, *Onboard Sensors* uses readings from some onboard sensors; both assume the auditor can access some of the onboard hardware, *Remote Motion Capture* and *Remote Camera Feed* use only external sensors. (Right) Tasks: two are navigation tasks, either velocity- or force-controlled, the other two are actuated joints tasks, including an Inverted Pendulum and a Legged Robot, either force- or torque-controlled.

(i) **Multi-Sine Wave**, inspired by replay attack detection (Ghamarilangroudi et al., 2025), embedding secret sinusoids detected via DFT energy; (ii) **Correlation-Based**, embedding a pseudo-random sequence detected via normalized cross-correlation; and (iii) **Tournament-Based**, a novel adaptation of SynthID (Dathathri et al., 2024) extended for continuous action spaces and remote robustness. The first three strategies handle synchronization uncertainty (C1) by maximizing the detection score over a grid of possible execution frequencies, while Tournament-Based does not require knowledge of the frequency and is thus not impacted by (C1). All approaches preserve the action distribution (W1). Full implementation and tuning details are in Appendix E.

We evaluate performance using three metrics, each estimated via batch bootstrapping (40 replications for remote glimpse modalities and 100 for others):

**Detectability.** As raw detection scores are not comparable across strategies, we assess how reliably watermarks are detected by comparing Receiver Operating Characteristic (ROC) curve. ROC give detection rates based on the relative score obtained by watermarked and non-watermarked policies.

**Anonymity.** A watermark should only be detectable by the intended owner. Strategies use an *owner key* $k$ for personalization, and detectors with the wrong key should fail. In other words, detectability should be high for the intended key $k$ and low for an incorrect key $k'$. Thus, denoting $\text{AUC}(k)$ the ROC Area Under Curve for key $k$, Anonymity is defined as $\text{Anonymity} = 1 - \text{AUC}(k')$, and higher Anonymity is better.

**Reward Preservation.** A watermark must preserve the original policy's performance. We evaluate this by comparing the reward distributions of watermarked and non-watermarked policies.

In the following, one replication refers to running replications of the watermarked policy and one replication of the non-watermarked policy. For each replication, we reset the policy, the environment, and the watermarking strategy, then generate a new signal of the given modality from scratch. The detection mechanism is applied independently to the outcome of each replication. To process *Remote Camera Feed* glimpses, we convert all camera feeds into velocity estimates using Template Matching from LuNežič et al. (2018). When experimenting with remote modalities in simulation, we use the raw rendered images as camera feed and discard all other data. Note that all the watermarking strategies considered in this work apply at inference time, meaning while the policy is deployed, and do not impact RL training. Thus, the different strategies can be deployed on the same pre-trained policies. In the following, we pre-train the same policies for all strategies and for each environment using Proximal Policy Optimisation (PPO) (Schulman et al., 2017).

## 7 EXPERIMENTAL RESULTS

To encode ownership, a watermarking strategy must preserve anonymity and receive a high detection score *only* when the auditor detects it with the secret key used during watermark generation. Our

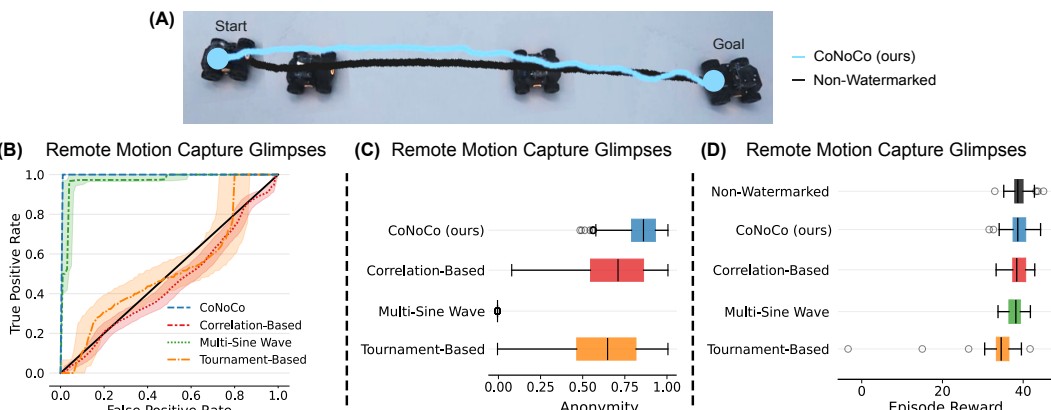

Figure 3: Results on the RoboMaster Navigation tasks. (A) Example trajectories of the watermarked and non-watermarked policies on the robot. (B) Detectability: ROC curve for 40 replications of the watermarked and non-watermarked policy for each baseline, lines indicate median and dashed areas quartiles. (C) Anonymity: computed as $1-$ area under the ROC curve, for detection with a different seed. (D) Reward Preservation: reward distribution of the watermarked and non-watermarked policies.

experimental results show that, among all the baselines we consider, only `CoNoCo` has this property. For example, we find that Multi-Sine-Wave has high detectability (as seen by its ROC curves), but really low anonymity, meaning that its watermark can be easily detected *with the wrong secret key*. The other baselines have better anonymity scores, but poor detectability, making them infeasible. Results for more environments are in Appendix I.

**RoboMaster Navigation.** We first demonstrate our approach in a real-world setting. We intentionally chose a simple task: navigating to a random 2D goal. This setting is challenging because behavioural redundancy is scarce and deviations are immediately visible, narrowing the margin for imperceptible modification. Success here emphasizes that `CoNoCo` is viable even in straightforward tasks, not just complex systems. We train a policy in simulation (VMAS (Bettini et al., 2022)) and deploy it on the RoboMaster platform (Blumenkamp et al., 2024), embedding the watermark online. We evaluate the *Remote Motion Capture* modality and use 40 replications, with each replication collecting 50s of data ($\approx 1000$ policy calls), and resetting the target upon arrival. The results in Fig. 3 show that `CoNoCo` consistently performs among the best in detection across modalities, while preserving anonymity and reward. The results for the same policy on the VMAS task in Appendix I.2 further show that while `CoNoCo` performs well in both simulation and real-world settings, other baselines degrade on the real robot. Crucially, `CoNoCo` succeeds despite the remote glimpse setup. The example trajectories (Fig. 3.A) confirm that the watermarked policy closely matches the non-watermarked one, showing that detectability does not induce visible behavioural changes. These results highlight the promise of `CoNoCo` for real-world robotics and remote detection.

**Force and Torque Controlled Tasks.** We next demonstrate generalization to more complex robotic systems, different control dynamics and *Remote Camera Feed* modality. As opposed to our real-world experiment, which used velocity commands, we watermark force-controlled policies in VMAS Navigation and Mujoco Inverted Pendulum (Todorov et al., 2012), and torque-controlled policies in Mujoco HalfCheetah (Brockman et al., 2016). All tasks use velocity estimates as glimpses, where the estimation methodology depends on the glimpse modality. We evaluate *Ground Truth Action*, *Onboard Sensors*, and *Remote Camera Feed* modalities (we omit *Remote Motion Capture* in simulation). For limbed robots such as HalfCheetah, the glimpses are the joint angular velocity, which we remotely estimate from a side-way camera feed by combining linear velocity estimates of the two extremities of each limb. We validate the watermarks across 100 replications. Data collected per replication: Navigation: 50s ($\approx 1000$ calls); Pendulum: 25s ($\approx 1000$ calls); HalfCheetah: 50s ($\approx 1000$ calls). Results are shown in Fig. 4. Across all tasks and glimpse modalities, `CoNoCo` achieves near-perfect detectability, despite the additional complexity of the tasks. This performance is only matched by Multi-Sine Wave, which, however, fails on anonymity. Importantly, `CoNoCo` also

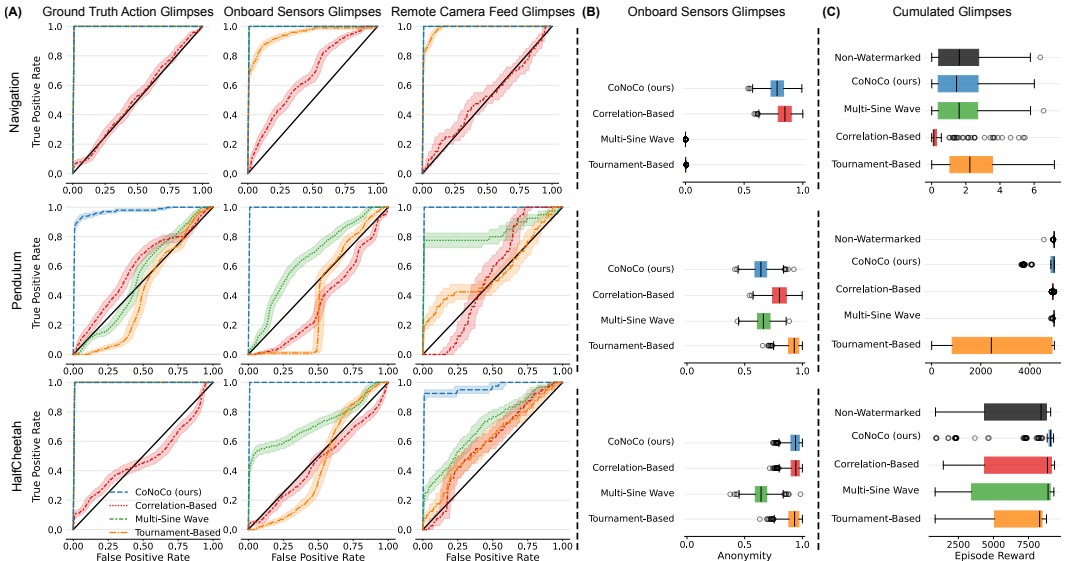

Figure 4: Results on a variety of Force and Torque Control tasks with increasing difficulty. (A) Detectability: ROC curve over 100 replications of the watermarked and non-watermarked policy for each baseline, lines indicate median and dashed areas quartiles. (B) Anonymity: computed as the complement to 1 of the ROC area under the curve for detection with a different owner seed, for *Onboard Sensors* glimpses. (C) Reward Preservation: reward distribution of the watermarked and non-watermarked policies.

obtains high detectability using *Remote Camera Feed*. This result highlights the promise of CoNoCo for more complex robotic tasks.

**Glimpse sequence length sensitivity.** Our experimental results in this section assume fixed glimpse sequence length. However, CoNoCo's detection quality correlates with glimpse sequence length, eventually converging on perfect detection (ROC AUC = 1). To understand how much data is required to reliably detect the watermark in our real robot experiments and each of our simulated environments, we examine CoNoCo's performance with respect to different glimpse sequence lengths. Our findings are presented in Appendix F.

**Adversarial Attacks.** Beyond the inherent challenges of remote detection, it is helpful for a watermarking scheme to be robust against deliberate attempts by an adversary to remove the signature. In Appendix H, we consider a number of such strategies, including white noise attacks, and jamming strategies specifically targeting the secret band $\mathcal{B}$. We find that CoNoCo is highly robust to such attacks: either they degrade policy performance, or they are unable to significantly harm detection. We also discuss (but do not experiment with) the feasibility of distillation as an adversarial attack.

## 8 CONCLUSION

Remotely detectable robot policy watermarking is an important capability for IP protection and safety certification in real-world robotics. We formalized the fundamental challenges posed by this type of watermarking, and proposed Colored Noise Coherency (CoNoCo), a robust, performance-preserving watermarking strategy designed for remote physical data. Our experiments, spanning different robot types and remote modalities, demonstrate that CoNoCo can successfully detect physical watermarks from remote data. Our results demonstrate how robot policy provenance can be verified non-invasively, paving the way for trustworthy deployment and accountability in real-world robot systems. We discuss open questions and limitations in Appendix A.

ACKNOWLEDGEMENTS

This work is supported by European Research Council (ERC) Project 949940 (gAIa), and by the EPSRC funded INFORMED-AI project EP/Y028732/1. We gratefully acknowledge their support.

REPRODUCIBILITY STATEMENT

We open-source our code at `https://github.com/proroklab/RobotPolicyWatermarking`, including the code used to train the policies, generate, inject, and detect the watermark across all tested modalities. The documentation linked in the README contains detailed instructions on how to reproduce the results and figures in this work. We also include the trained policies themselves, to enable exact replication of our simulation results. We designed this codebase to be straightforward to install and use, so that new users can easily implement new watermarking strategies and test them on our environments to try and beat our proposed approach.

All mathematical theorems claimed in this work are proven in the Appendix.

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

# A    OPEN QUESTIONS AND LIMITATIONS

As shown in our theoretical analysis and experiments, the CoNoCo watermark is robust to different policy behaviours, glimpse modalities, and robot types (e.g., legged or wheeled). Nevertheless, it is limited by the quality of the glimpse data it receives, which may be even less clean in real-world settings than in our experiments. In Appendix G, we provide an empirical study of the sensitivity of detection to various forms of glimpse alteration. In all our experiments, we could record the robots' movements with a stationary side-view or top-view camera that had full visibility at all times. In many real-world scenarios, however, robots may be partially obscured due to other moving bodies, camera angles or self-occlusion. Such setups would make it challenging to extract reliable motion glimpse estimates. Addressing this limitation would require more advanced computer vision techniques, which are beyond the scope of this work and are left for future study. Addressing these challenges will significantly broaden the applicability of CoNoCo in unpredictable, real-world domains.

CoNoCo is designed around policy stochasticity, which provides the variability needed to embed and recover the watermark. Applying the method to deterministic policies would require adapting these ideas to alternative sources of variability. We view this as a natural and promising extension of our framework and leave it for future work.

# B    USE OF LLMS

We used LLMs (Gemini and ChatGPT) to discover new references related to our work, as well as polish some parts of the writing. We read all suggested references ourselves and verified their relevance.

# C    CoNoCo IN LTV SYSTEMS

Real robotic systems often deviate from the idealized LTI (Linear Time-Invariant) assumptions, presenting as LTV (Linear Time-Varying) systems with changing dynamics $\mathcal{S}_{dyn}(t)$ and exploration scale $\Sigma_k$. This presents two main challenges and corresponding mitigation strategies employed by CoNoCo.

## C.1    TIME-VARYING EXPLORATION SCALE ($\Sigma_k$)

**The Challenge: Spectral Smearing.** The watermark $W_k$ is scaled by the policy's exploration scale $\Sigma_k$. If $\Sigma_k$ changes rapidly (e.g., the robot suddenly switches from cautious exploration to decisive movement), it modulates the amplitude of the watermark. This modulation (like rapidly changing the volume of a specific tone) spreads the energy of $W_k$ outside the secret band $\mathcal{B}$. This effect, known as "spectral smearing," reduces the detectable signal energy within the band, lowering the SINR and making detection harder.

**Mitigation.** CoNoCo works best when $\Sigma_k$ evolves slowly. We find it robust to this effect empirically, achieving high detection rates despite varying exploration scales in all our experiments. However, if $\Sigma_k$ evolves abruptly, a potential mitigation strategy (which we do not employ in this work) is to replace $\Sigma_k$ with a *moving average* of the last several exploration scales: $\overline{\Sigma}_k$. This smooths the modulation, reducing spectral smearing and improving detection. This introduces a trade-off: larger averaging windows improve detection but may slightly impact policy performance if the responsiveness of the exploration scale is critical.

## C.2    LTV DYNAMICS AND ESTIMATION CHALLENGES

**The Challenge: Phase Variation and Estimation Bias.** Coherency is formally defined for LTI systems, where the relationship (including the timing, or phase) between input and output is constant. CoNoCo analyzes the signals of real-world LTV systems using Short-Time analysis (implemented via Welch's method), which divides the signal into short windows. If these dynamics change over time, particularly the phase relationship between the input (watermark) and the output (glimpse), this averaging can lead to cancellation (destructive interference) of the Cross-Spectral Density (CSD).

This cancellation biases the coherency magnitude estimate downwards, making the watermark harder to detect. This is a known limitation when analyzing LTV systems.

**Mitigation.** CoNoCo works best when the system dynamics do not change over time. In particular, if the system is approximately LTI within each window, the detection signal will be stronger. However, CoNoCo also has a number of mitigation elements that improve its robustness when this is not the case:

(i) **Multi-Dimensional Averaging (Spatial Diversity).** The final detection score (Section 4) is calculated by averaging the *magnitude* of the different physical dimensions $D$. Complex robots are typically monitored through multiple sensors or observation dimensions (e.g., different joint angles, velocities, or viewpoints in a camera feed). It is unlikely that all dimensions are affected equally by time-varying dynamics. Some physical dimensions may behave much more linearly (LTI-like) than others. By averaging the detection scores across all available dimensions, CoNoCo exploits this spatial diversity. Strong detection signals from the well-behaved, more linear dimensions can ensure successful watermarking, even if other dimensions provide a weaker signal due to strong LTV effects. Furthermore, while not explored in this work, one could potentially improve detection further by identifying and censoring dimensions that exhibit highly non-linear behavior.

(ii) **Strategic Band Selection ($\mathcal{B}$).** The design of CoNoCo allows the owner to choose the secret frequency band $\mathcal{B}$. This band can be strategically selected to target frequencies where the robot's physical response is known to be relatively stable, predictable, and linear (more LTI-like). Conversely, we can avoid frequencies associated with highly unstable or rapidly changing dynamics (e.g., resonant frequencies or behaviors involving abrupt contact changes) where the LTV effects are most pronounced (see Appendix D).

## D TUNING CoNoCo

In general, we find that CoNoCo is fairly robust and works "out of the box" when given sufficiently long glimpse sequences. However, in cases where the glimpse sequences are very short, or in rare cases where we observe that policy performance is impacted by the autocorrelation introduced by the watermark injection, detection and performance can be improved by tuning the frequency band $\mathcal{B}$ and the window length of Welch's method $T_{win}$.

$\mathcal{B}$ should be selected to not interfere with the policy performance and ideally, interact with the system dynamics and policy as little as possible: this might mean omitting low frequencies (if the policy varies smoothly), or high frequencies (if the policy varies highly and requires precision). $T_{win}$ should be selected based on the length of the glimpse sequence. When $f_g/f_\pi = 5$ and we observe 1000 policy calls–so the glimpse sequence has length 5000–we find $T_{win} = 64$ to work well. When the glimpse sequence is of length 20000 or more, $T_{win} = 256$ works best. For intermediate values, $T_{win} = 128$ may work.

In the case of the VMAS Velocity Navigation environment, we found that the glimpse sequence length of 5000 (1000 policy steps at a glimpse-to-policy-call ratio of 5) meant that our default window of $T_{win} = 256$ was too large. Setting the window to $T_{win} = 64$ attained almost perfect detection results.

In the case of the Pendulum environment, we found that using the band $\mathcal{B} = [1.2, 2.49]$ was interfering with the policy performance (but not watermark detection). Reducing the higher frequencies by setting $\mathcal{B} = [0.1, 1.5]$ made the policy perform equivalently to its non-watermarked variant while not harming detection.

## E BASELINE WATERMARKING STRATEGIES

To the best of our knowledge, our proposed watermarking strategy is the first that enables remote detection. As no direct baseline exists for this setting, we introduce adapted variants of prior watermarking methods, which, although not originally designed for remote detection, serve as the most relevant points of comparison.

**Multi-Sine Wave.** Inspired by techniques used for replay attack detection (Ghamarilangroudi et al., 2025), this strategy embeds a watermark by synthesizing a signal composed of a sum of multiple

sinusoids. The owner key defines the secret frequencies and signs of these sinusoids within a specific band. This synthesized signal, normalized to preserve the policy's action distribution (W1), replaces the standard exploration noise. Detection is performed in the frequency domain. The detector calculates the energy of the observed glimpses at the secret frequencies using a Discrete Fourier Transform (DFT). The detection score is the signed sum of these energies, maximized over a search grid of possible policy execution frequencies to address synchronization uncertainty (C1).

**Correlation-Based.** This strategy embeds a watermark by replacing the policy's exploration noise with a secret pseudo-random sequence. For detection, the glimpses are first high-pass filtered to isolate the watermark signal from the primary behaviour (C3). The detector then calculates the normalized cross-correlation between the filtered glimpses and the hypothesized watermark signal, maximizing this score over a range of possible policy execution frequencies to handle synchronization uncertainty (C1).

**Tournament-Based.** SynthID (Dathathri et al., 2024) is a tournament-based method for watermarking that represents the state of the art in watermarking LLMs. However, the token generation setting used in SynthID differs from ours in two respects: (1) the support of the token distribution is discrete, and (2) detection is not remote, as the exact LLM output is always available to the detector. We therefore introduce a variant, which we term *Tournament-Based*, designed to: (1) extend to distributions with continuous support, and (2) enable remote detection by (2.i) ensuring robustness to noise through assigning similar scores to neighbouring actions, and (2.ii) relying exclusively on information available in the glimpses when selecting watermarked actions, so it can be inverted for detection. One round of Tournament-Based proceeds as follows. First, $N$ actions are sampled from the policy distribution for the current timestep, which enables (1). These $N$ actions encounter a tournament, where the winner of each duel is determined using scoring functions known as g-functions. Here, to enforce property (2.i), we use Bell-shaped g-functions, with the parameters of the Bell-curve sampled using a context-dependent random key. To enforce property (2.ii), we choose this random key as the norm of the observation that will later be available through glimpses (e.g. velocities). The owner key is used to build the continuous function that is sampled to get the g-value parameters.

We tuned all of these watermarks, including `CoNoCo`, by rerunning our experimental suite a few dozen times on the Velocity-command VMAS Navigation (Appendix I.2) and a small handful of times on the other environments. We updated the parameters by hand after each attempt, optimizing for detection AUC. In all methods except the tournament-based method, a hyperparameter of interest is the granularity of the search for the hypothesized true frequency $f_{pi}$ (see Algorithm 1). For Multi-Sine Wave, the additional relevant parameters are the band $\mathcal{B}$ (similar to `CoNoCo`), the number of sinusoids. To tune $\mathcal{B}$, we followed the same recommendations as for `CoNoCo`, described in Appendix D. For the tournament-based method, the choice of $g$ function and the number of tournament layers was tuned.

Full implementation and hyperparameter details are available in our code repository.

# F  GLIMPSE SEQUENCE LENGTH SENSITIVITY

In Figure 5, we analyze how the number of timesteps in the glimpse sequence influences watermark detectability across environments. Detectability is measured using the Area Under the ROC Curve, where a value of 1.0 indicates perfect classification. This analysis quantifies the amount of data necessary for reliable detection of the watermark in each environment. Across most environments, detectability saturates at roughly 1000 timesteps. These results determine the glimpse length used in our main experiments, ensuring that all evaluations operate in a regime of near-maximal detectability.

# G  GLIMPSE ALTERATION SENSITIVITY

This section studies the sensitivity of `CoNoCo` to various glimpse alterations to assess its robustness.

## G.1  ROBUST OFFSET SYNCHRONIZATION VIA GCC-PHAT

An important challenge in remote auditing (C1) is the unknown time offset: the auditor may begin observing the system (e.g., via CCTV footage) long after the robot started operating. The standard `CoNoCo` detection algorithm (Algorithm 1) assumes the glimpse sequence $G$ starts near the beginning

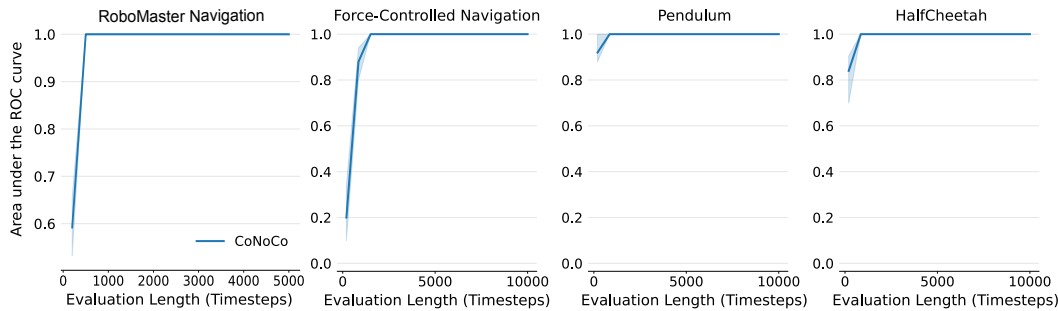

Figure 5: Relationship between *glimpse sequence length* and the watermark detectability of `CoNoCo`. Detectability is reported as the ROC AUC averaged over 10 repetitions. We use the Onboard Sensors glimpse modality, except for "RoboMaster Navigation," where we instead use real-world data from our robot experiments with Motion Capture glimpses. Shaded regions indicate quartiles.

of the policy execution (Line 7: $W'_d \leftarrow W'_s[0 : |G|, d]$). If there is a large offset $\tau$, the alignment is lost, causing detection to fail.

This can be robustly handled with a simple change to `CoNoCo`, by the Generalized Cross-Correlation Phase Transform (GCC-PHAT) (Knapp & Carter, 2003) to estimate and correct for the time offset $\tau$. GCC-PHAT is a technique for estimating the time delay between two signals. Standard cross-correlation is sensitive to system dynamics (C2), as the robot's physics distort the signal amplitude. In contrast, GCC-PHAT makes the synchronization process invariant to Linear Time-Invariant (LTI) system dynamics, mirroring the invariance property of Spectral Coherency (Theorem 5.2). As GCC-PHAT is well-known, we refer to (Knapp & Carter, 2003) for the details.

**Implementation.** For each frequency hypothesis $s$, we calculate the GCC-PHAT curve between the glimpses $G$ and the resampled watermark $W'_s$ within the secret band $\mathcal{B}$. For MIMO systems, we aggregate these curves across dimensions (Lines 6-10) to find a robust consensus offset $\hat{\tau}$ (Line 11). Finally, the coherency score is calculated on the synchronized segments (Lines 12-19).

---

**Algorithm 2** Watermark detection with offset handling via GCC-PHAT (`CoNoCo-Offset-Handling`)

---

**Require:** Glimpses $G$, Glimpse Freq $f_g$, Key $\mathcal{K} = \{S, \mathcal{B}\}$, Search Range $\mathcal{F}_{\text{search}}$, Max Offset $T_{max}$
1: $D \leftarrow \text{NumDimensions}(G)$; $BestScore \leftarrow 0$
2: $W_{base} \leftarrow \text{RegenerateWatermarkSequence}(\dots, \text{Length} \approx |G|/f_g + T_{max})$
3: **for** $s \in \mathcal{F}_{\text{search}}$ **do**
4:     $W'_s \leftarrow \text{ResamplePoly}(W_{base}, f_g/s)$
5:     $CCC_{agg} \leftarrow 0$                                                            $\triangleright$ Offset handling starts here
6:     **for** $d = 1$ to $D$ **do**
7:         $CCC_d \leftarrow \text{CalculateGCCPHAT}(G[:, d], W'_s[:, d], f_g, \mathcal{B})$
8:         $CCC_{agg} \leftarrow CCC_{agg} + CCC_d$
9:     $\hat{\tau} \leftarrow \text{argmax}(CCC_{agg})$   $\triangleright$ Find consensus offset across the D dimensions. Offset handling ends here!
10:     $W'_{\text{aligned}} \leftarrow W'_s[\hat{\tau} : \hat{\tau} + |G|]$               $\triangleright$ Rest of the detection algorithm remains the same.
11:     $Score_{sum} \leftarrow 0$
12:     **for** $d = 1$ to $D$ **do**
13:         $f, C_d \leftarrow \text{Coherency}(G[:, d], W'_{\text{aligned}}[:, d], f_g)$
14:         $Score_d \leftarrow \text{Mean}(\text{Abs}(C_d[f \in \mathcal{B}]))$
15:         $Score_{sum} \leftarrow Score_{sum} + Score_d$
16:     **if** $Score_{sum}/D > BestScore$ **then**
17:         $BestScore \leftarrow Score_{sum}/D$
18: **return** $BestScore$

---

**Empirical Validation.** The results in Figure 6 demonstrate the effectiveness of this enhancement. In all experiments, standard `CoNoCo`'s performance degrades rapidly as the offset increases, whereas `CoNoCo-Offset-Handling` maintains near-perfect detection (AUC $\approx 1.0$) even for substantial

offsets. This confirms that `CoNoCo` can be readily adapted for realistic auditing scenarios involving arbitrary observation start times.

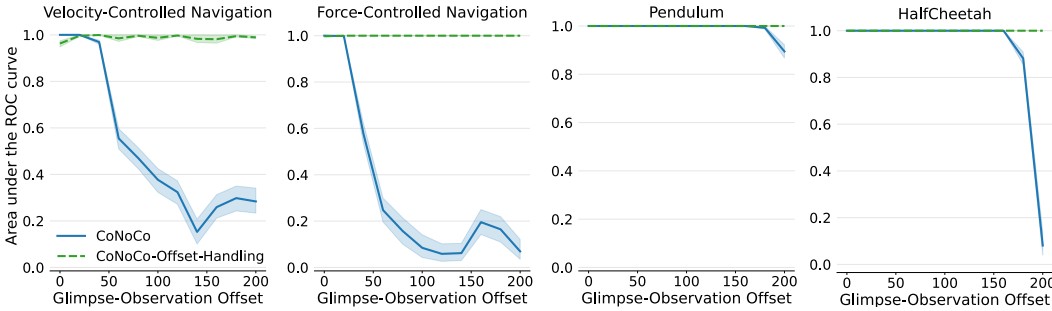

Figure 6: Relationship between *glimpse offset* and the detectability of `CoNoCo` (Algorithm 1) and `CoNoCo-Offset-Handling` (Algorithm 2). Detectability is reported as the ROC AUC over 100 repetitions, using the Onboard Sensors glimpse modality. Shaded regions indicate quartiles.

### G.2 ROBUTNESS TO TIME-JITTER

Another important challenge in real-world glimpses is time-jitter, which corresponds to inconsistent intervals between glimpses within a sequence. Such jitter can arise either at the policy level, where the policy is queried at slightly inconsistent time intervals (due to hardware constraints, network traffic, etc.), or at the glimpse-acquisition level, where the sensor captures glimpses at slightly inconsistent intervals (due to the properties of the specific sensor at hand). We measure jitter as the relative standard deviation of the glimpse interval. For example, a jitter of $0.2$ for glimpses at frequency $100Hz$ means that the interval between two glimpses is drawn from a normal distribution with mean $1/100 = 0.01s$ and std $0.2/100 = 0.02s$.

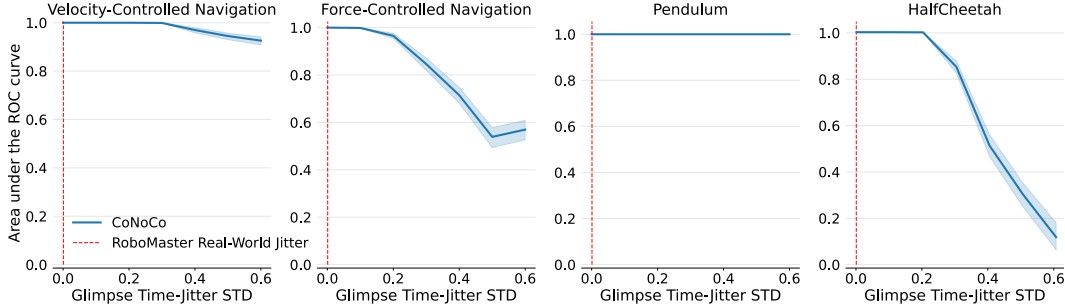

Figure 7: Relationship between *glimpse time-jitter* and the detectability of `CoNoCo`, reported as the ROC AUC over 100 repetitions, using the Onboard Sensors glimpse modality. Shaded regions indicate quartiles. The vertical red line denotes the actual time jitter quantified in our experiments on the RoboMaster platform.

We study the effect of increasing jitter on `CoNoCo` in Figure 7. For comparison, we quantify the time jitter in the Remote Motion Capture Glimpses collected on RoboMaster for our main experimental results and find a value of $0.002$, which we include as a reference in the figure. The results demonstrate that `CoNoCo` remains robust under levels of jitter significantly higher than those observed in our real-world system.

### G.3 ROBUSTNESS TO DROP OF GLIMPSES

Another alteration that can impact detection in real-world settings is glimpse-drop, where a proportion of the glimpses is missed at random point during detection. We study the sensitivity of `CoNoCo` to such drop by removing an increasing number of randomly-sampled glimpses across tasks. The results in Figure 8 show that `CoNoCo` remains robust to glimpse-drop up to $200$ frames, which represent $20\%$ of frames in the Navigation and Pendulum tasks and $5\%$ in the HalfCheetah tasks.

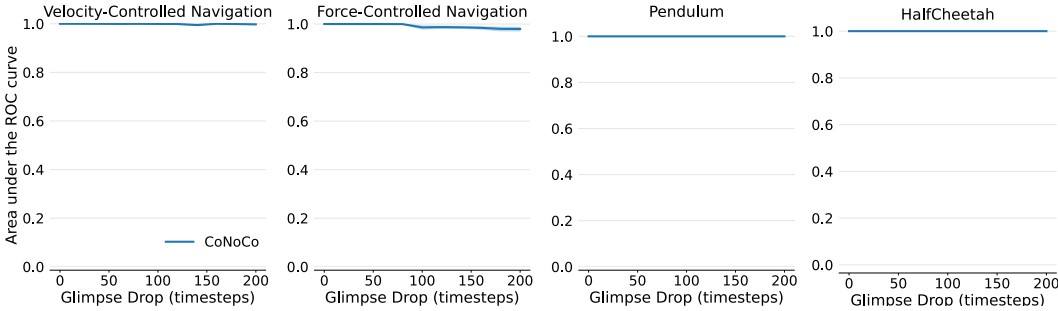

Figure 8: Relationship between *glimpse drop* and the detectability of `CoNoCo`, reported as the ROC AUC over 100 repetitions, using the Onboard Sensors glimpse modality. Shaded regions indicate quartiles.

### G.4 ROBUSTNESS TO REMOTE GLIMPSES SHIFT

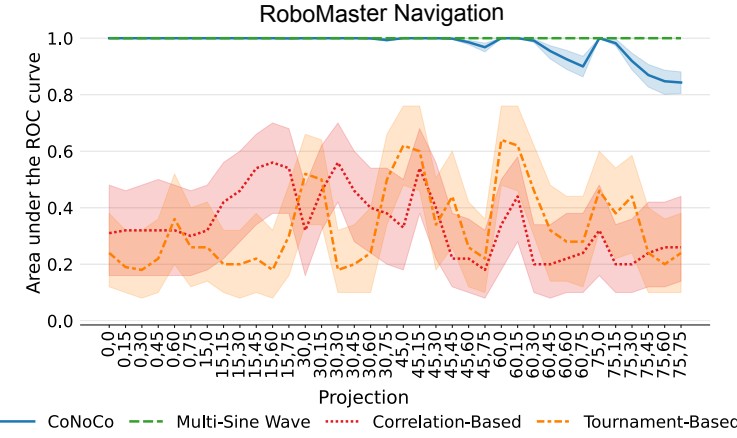

Figure 9: Relationship between *Motion Capture Glimpses Shift Angle* and the detectability of `CoNoCo`. Angles are specified as tuples of the x and y axes rotations (in degrees). Detectability is reported as the ROC AUC over 100 repetitions, using the Motion Capture Glimpses modality. Shaded regions indicate quartiles.

When using remote auditing (C1), another potential challenge in real-world scenarios is that the camera or sensor may be slightly shifted relative to the side-view or top-down view adopted in our experimental setup. We study the robustness of `CoNoCo` to this type of alteration by projecting the glimpses onto planes that are slightly shifted with respect to the original one. In Figure 9, we present this analysis for the real-world RoboMaster data and report how the ROC AUC evolves as the angle between the original and projected planes increases. We report the angles as tuples of two elements, containing the rotations along the x and y axes with respect to the original glimpse plane. Our results highlight that the detection performance of `CoNoCo` is only affected for very large angular deviations from the original plane (greater than $60°$ along both x and y), indicating strong robustness to such shifts.

## H ADVERSARIAL ROBUSTNESS ANALYSIS

We consider a threat model where the adversary is the Policy User (Figure 1, Step 2), aiming to evade detection while preserving the utility of the stolen policy $\tilde{\pi}_\theta$. We assume the adversary does not possess the secret key $\mathcal{K}$.

We first consider robustness to additive noise. Then, because `CoNoCo` uses Colored Gaussian Noise (CGN), it alters the spectral distribution of the actions. We acknowledge that a sophisticated adversary analyzing long-term spectral averages might be able to localize the secret band $\mathcal{B}$. We analyze the

strategies the adversary might employ if they identify the band. We show that learning the band is generally unhelpful. We finally discuss distillation / behaviour cloning as a possible vehicle for IP theft.

## H.1 ROBUSTNESS TO ADDITIVE NOISE ATTACKS

**Additive Noise Attack (Randomized Smoothing).** We consider the effects of an *Additive Noise Attack*. The adversary adds White Gaussian Noise (WGN) to the actions before execution:

$$a'_k = \text{Clip}\left(a_k + \eta_{adv}\right), \tag{1}$$

where $\eta_{adv} \sim \mathcal{N}(0, \sigma^2_{adv} I)$, and the clipping ensures the actions remain within the environment's physical bounds. The standard deviation $\sigma_{adv}$ represents the adversary's strength or budget.

This attack directly impacts detectability by introducing an additional source of noise into the system. Referring to the analysis in Section 5, the addition of $\eta_{adv}$ increases the overall noise power $P_N(f)$, consequently reducing the Signal-to-Interference-plus-Noise Ratio (SINR) (Definition 5.1). According to Theorem 5.3, this directly lowers the expected magnitude of the coherency, making detection more difficult.

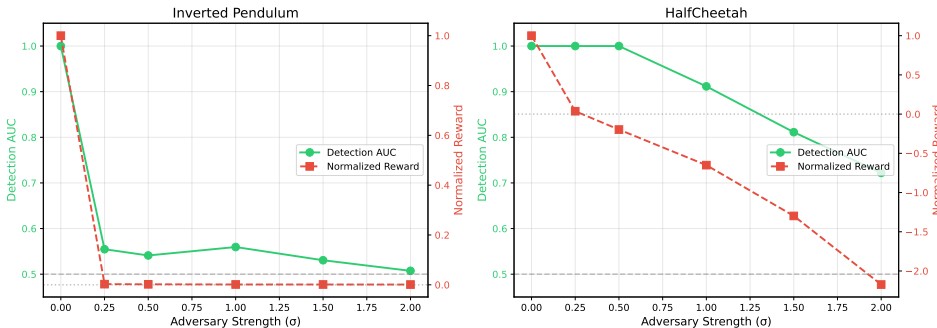

Figure 10: Adversarial robustness results for additive noise attacks (results over 10 repeats). Each subfigure shows detection scores and *normalized* policy reward as a function of adversarial noise strength $\sigma_{adv}$ ranging from 0 to 2. Normalized reward means the ratio of reward at each adversary strength to the baseline reward. The plots demonstrate CoNoCo's resilience, with detection persisting under high noise in HalfCheetah, while requiring severe reward degradation to evade detection in Inverted Pendulum.

**Evaluation and Results.** To evaluate CoNoCo's robustness, we simulate this attack on the Inverted Pendulum and HalfCheetah tasks, varying the adversarial strength $\sigma_{adv}$ from 0 to 2. We analyze the trade-off between the reduction in the detection scores and the degradation in the policy's reward. The results, shown in Figure 10, demonstrate that CoNoCo is strongly robust to this type of adversarial attack. In the HalfCheetah task, both detection AUC and policy reward degrade gradually as $\sigma_{adv}$ increases, but the watermark remains consistently identifiable up to $\sigma_{adv} = 2$. This is particularly robust given that, at $\sigma_{adv} = 2$, the added adversarial noise overwhelms the policy's original actions, yet detection persists. In contrast, for the Inverted Pendulum task, even a modest $\sigma_{adv} = 0.25$ severely degrades *both* detection and reward, indicating that the attack cannot evade watermarking without rendering the policy ineffective. Overall, these findings show that additive noise attacks perform poorly against CoNoCo, as evading detection requires noise levels that destroy the policy's value, demonstrating its robustness.

## H.2 ADVERSARIAL FILTERING ATTACK (BAND-STOP)

If the adversary has successfully learned $\mathcal{B}$, a direct attack is to apply a band-stop (notch) filter $H_{stop}$ to the actions $a'_k = H_{stop}(\tilde{a}_k)$. While this successfully attenuates the watermark $W_k$, the filter is applied to the *entire* action, meaning it also removes components of the original policy behavior $\mu_k$ within $\mathcal{B}$.

This acts as **self-jamming**. If the band $\mathcal{B}$ contains frequencies important for the robot's task performance, this filtering degrades the utility of the stolen policy.

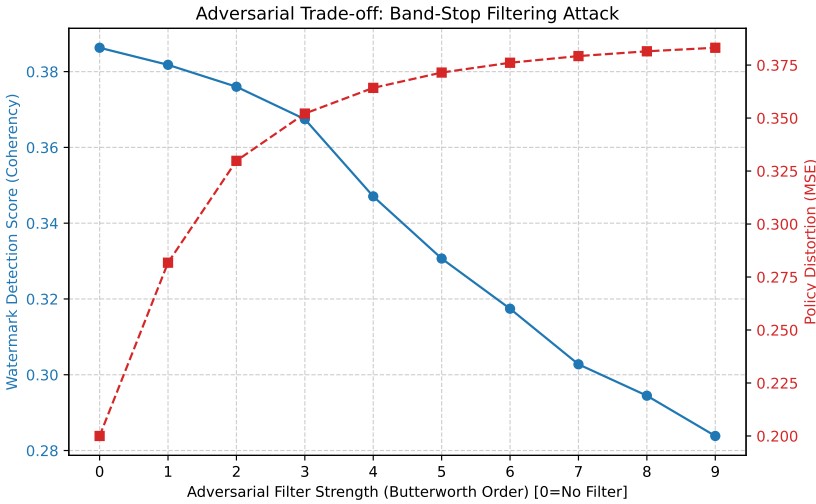

Figure 11: Adversarial Trade-off for Band-Stop Filtering Attack. As the adversary increases the filter strength (Order), the watermark detection score (Blue) decreases, but the distortion (MSE) to the original policy behavior (Red) increases dramatically (by 257% in this simulation), demonstrating the degradation of policy utility.

We demonstrate this trade-off in a numerical experiment (Figure 11), where the policy behaviour partially intersects the secret band. Roughly speaking, if $x\%$ of the frequency content of a policy $\mu_k$ lies inside the secret band $\mathcal{B}$, and we apply a filter to completely dampen this band, the distortion (Mean Squared Error) between the intended policy behaviour $\mu_k$ and the executed action $a'_k$ will be $x\%$, harming performance.

To demonstrate this, we consider a synthetic policy $\mu_k$, generated from colored Gaussian noise with $60\%$ of its signal intersecting the secret band $\mathcal{B}$, and $40\%$ of its signal intersecting another, arbitrary frequency band. As the filter strength increases, the watermark detection score drops, but the distortion (Mean Squared Error) between the intended behavior $\mu_k$ and the executed action $a'_k$ increases significantly. In our simulation, applying a strong filter increased policy distortion to $37.5\%$ relative to the baseline. Since almost all policies will intersect with the secret band $\mathcal{B}$ to some extent, this demonstrates that band-stop adversarial attacks against CONOCO come at a significant performance cost, defeating the point of such adversarial methods.

## H.3 RL-BASED STRUCTURED JAMMING

One might consider an approach wherein some RL agent learns a structured jamming signal $J_k$ to interfere with the watermark. Optimal interference implies *cancellation* ($J_k \approx -\Sigma_k W_k$). However, this is rigorously impossible without the secret key. Cancellation requires predicting the instantaneous value (phase) of the pseudo-random signal $W_k$. Since the adversary does not know the key (the seed for $W_k$), $W_k$ and the learned jamming signal $J_k$ are independent, zero-mean stochastic processes. Mathematically, the variance (power) of the sum is the sum of the variances: $Var(W + J) = Var(W) + Var(J)$. The power always increases.

We verified this numerically and visualize it using Power Spectral Density (PSD) in Figure 12. The combined signal power (Red) is the sum of the individual powers (Blue and Green), confirming that the signals add rather than cancel. Therefore, the adversary cannot implement structured cancellation. Their best strategy reverts to the Additive Noise Attack analyzed in Section H.1.

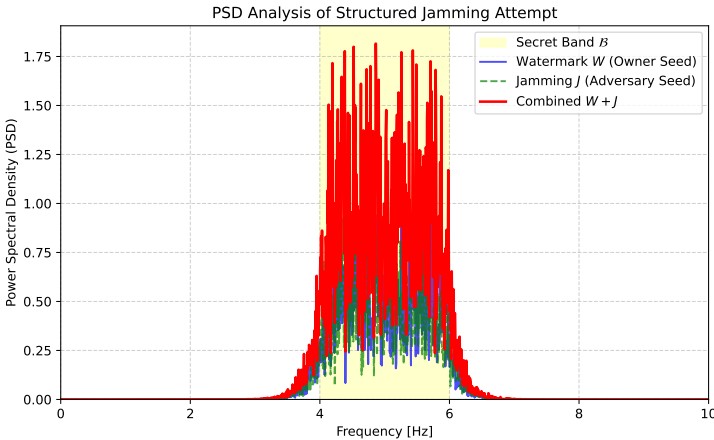

Figure 12: PSD Analysis of Structured Jamming Attempt. The Watermark (W) and Jamming signal (J) are generated with different seeds. The power of the combined signal (W+J) is the sum of the individual powers, confirming that cancellation is impossible without the secret key.

### H.4    POLICY DISTILLATION

Policy distillation might remove the watermark as a side-effect of the imperfect approximation. While potentially effective, we outline two possible challenges when deploying such approaches:

1. IP-sensitive policies, such as foundation models and autonomous driving policies, are often trained on vast amounts of diverse and proprietary real-world data (sometimes gathered from different robots and task types) and synthetic (simulator) data. It is not necessarily feasible or cost-effective for our adversary to distil the original policy purely by collecting data from deploying such policies on a single robot, a handful of robots, or even an API (which may include rate limits or incur significant usage costs at the scale required for distillation).

2. Building on the above, since distillation is an imperfect approximation, it often leads to a performance drop, reducing the value of the stolen IP (e.g., Cho & Hariharan (2019)).

We conclude that, while distillation may be an effective method against our watermark, it requires specific conditions to be an attractive adversarial method against `CoNoCo`.

## I    RESULTS ON ADDITIONAL ENVIRONMENTS

### I.1    RESULTS ON FORCE-CONTROLLER VMAS NAVIGATION WITH OBSTACLES

We introduce an additional environment: Force-Controller VMAS Navigation with dynamically resampled obstacles. In this setting, the agent navigates among eight obstacles whose positions are randomly resampled each time the agent reaches the goal, resulting in multiple reconfigurations per episode. We choose this environment because its cluttered layout intrinsically constrains the policy's exploration noise during certain phases of the task. Our aim is to evaluate whether `CoNoCo` remains effective under such limited-variance conditions. The results, shown in Figure 13, indicate that CoNoCo maintains strong detection performance, demonstrating its robustness even in low-variance environments.

### I.2    RESULTS ON VELOCITY-CONTROLLER VMAS NAVIGATION

We provide in Figure 14 the results for the Velocity-Controlled VMAS Navigation task. The policy used here is also the one deployed on the RoboMaster in the main results. The results show that baselines that got low detectability on the real robot are getting better results on the simulated task. `CoNoCo` is highly successful in both the real and simulated task.

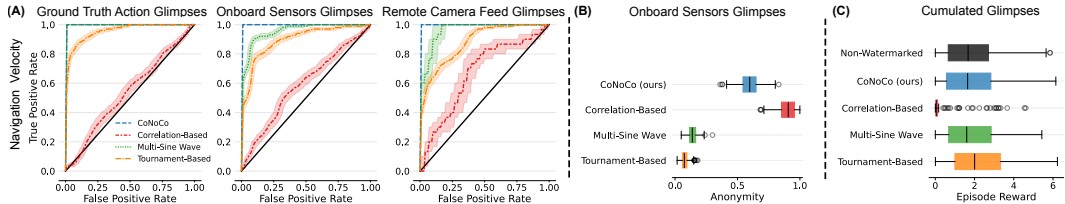

Figure 13: Results on the Velocity-Controlled VMAS Navigation with Obstacles task. A) Visualization of the environment. The blue dot represents the agent, the black dots denote the obstacles, and the line shows an example trajectory. (B) Detectability: ROC curve for $100$ replications of the watermarked and non-watermarked policy for each baseline, lines indicate median and dashed areas quartiles. (C) Anonymity: computed as the complement to $1$ of the ROC area under the curve for detection with a different owner seed, for *Onboard Sensors* glimpses. (D) Reward Preservation: reward distribution of the watermarked and non-watermarked policies.

Figure 14: Results on the Velocity-Controlled VMAS Navigation task. T(A) Detectability: ROC curve for $100$ replications of the watermarked and non-watermarked policy for each baseline, lines indicate median and dashed areas quartiles. (B) Anonymity: computed as the complement to $1$ of the ROC area under the curve for detection with a different owner seed, for *Onboard Sensors* glimpses. (C) Reward Preservation: reward distribution of the watermarked and non-watermarked policies.

## J   PROOF OF THEOREM 5.1 (MARGINAL DISTRIBUTION PRESERVATION)

*Proof of Theorem 5.1 (Marginal Distribution Preservation).* Let $H$ be a stable LTI filter with impulse response $\{h_j\}$. The input is a sequence of independent and identically distributed (i.i.d.) White Gaussian Noise $X_k \sim \mathcal{N}(0,1)$. The raw filtered output $W_k^{\text{raw}}$ is given by the convolution:

$$W_k^{\text{raw}} = (H * X)_k = \sum_{j=-\infty}^{\infty} h_j X_{k-j} \tag{2}$$

**1. Gaussianity:** Since the input variables $X_{k-j}$ are jointly Gaussian (due to independence), and $W_k^{\text{raw}}$ is a linear combination of these variables, $W_k^{\text{raw}}$ is also a Gaussian random variable.

**2. Mean:** We calculate the expected value using the linearity of expectation:

$$E[W_k^{\text{raw}}] = E\left[\sum_j h_j X_{k-j}\right] = \sum_j h_j E[X_{k-j}] \tag{3}$$

Since $E[X_k] = 0$ for all $k$, we have $E[W_k^{\text{raw}}] = 0$.

**3. Variance:** We calculate the variance. Since the mean is zero, $Var[W_k^{\text{raw}}] = E[(W_k^{\text{raw}})^2]$.

$$Var[W_k^{\text{raw}}] = E\left[\left(\sum_j h_j X_{k-j}\right)\left(\sum_m h_m X_{k-m}\right)\right] \tag{4}$$

$$= \sum_j \sum_m h_j h_m E[X_{k-j} X_{k-m}] \tag{5}$$

Since $X_k$ is WGN with unit variance, $E[X_n X_m] = \delta_{nm}$ (Kronecker delta). The expectation is non-zero only when $j = m$.

$$Var[W_k^{\text{raw}}] = \sum_j h_j^2 = \sigma_{W^{\text{raw}}}^2 \tag{6}$$

This sum converges because the filter $H$ is stable.

**4. Normalization:** The final watermark sequence $W_k$ is normalized by the standard deviation:

$$W_k = \frac{W_k^{\text{raw}}}{\sigma_{W^{\text{raw}}}} \tag{7}$$

$W_k$ remains Gaussian with mean $E[W_k] = 0$. Its variance is:

$$Var[W_k] = \frac{Var[W_k^{\text{raw}}]}{\sigma_{W^{\text{raw}}}^2} = 1 \tag{8}$$

Therefore, the marginal distribution of $W_k$ is $\mathcal{N}(0, 1)$. $\qquad\square$

## K    PROOF OF THEREOM 5.2 (INVARIANCE OF COHERENCY MAGNITUDE UNDER LTI FILTERING)

*Proof.* We have $S_{YY}(f) = |H_{sys}(f)|^2 S_{XX}(f)$ and $S_{XY}(f) = H_{sys}(f)^* S_{XX}(f)$. This implies $|C_{XY}(f)| = \frac{|S_{XY}(f)|}{\sqrt{S_{XX}(f) S_{YY}(f)}} = \frac{|H_{sys}(f)^* S_{XX}(f)|}{\sqrt{S_{XX}(f) \cdot |H_{sys}(f)|^2 S_{XX}(f)}} = 1$. $\qquad\square$

## L    PROOF OF THEOREM 5.3 (COHERENCY AND SINR)

*Proof.* We analyze the system under the LTI assumptions stated in the theorem. The input to the dynamics $H_{sys}$ is the action $\tilde{a} = \mu + \Sigma W$. The observed glimpse $G$ is the output of the dynamics plus sensor noise $\eta$. We aim to calculate the magnitude squared coherency between the watermark $W$ and the glimpse $G$, $|C_{WG}(f)|^2$.

Due to the linearity of the system, the glimpse $G$ (in the time domain) is a superposition of the responses:

$$G(t) = (\Sigma H_{sys} * W)(t) + (H_{sys} * \mu)(t) + \eta(t). \tag{9}$$

We decompose $G(t)$ into two components: $G(t) = S(t) + N(t)$.

The signal component $S(t)$ is the part derived from the watermark $W$:

$$S(t) = (\Sigma H_{sys} * W)(t). \tag{10}$$

$S(t)$ is the output of an LTI system (defined by the combined response $\Sigma H_{sys}$) with input $W(t)$.

The noise/interference component $N(t)$ includes the policy interference and sensor noise:

$$N(t) = (H_{sys} * \mu)(t) + \eta(t). \tag{11}$$

By assumption, $W, \mu, \eta$ are mutually independent. Therefore, the input $W(t)$ is independent of the noise component $N(t)$. Furthermore, the signal $S(t)$ (derived only from $W$) is independent of $N(t)$.

We define the signal power $P_S(f)$ and noise power $P_N(f)$ in the glimpse $G$ as the PSDs of $S(t)$ and $N(t)$ respectively (Definition 5.1):

$$P_S(f) = S_{SS}(f) \tag{12}$$
$$P_N(f) = S_{NN}(f) \tag{13}$$

Since $G = S + N$ and $S$ and $N$ are independent (and thus uncorrelated, assuming standard zero-mean processes for spectral analysis), the PSD of the glimpse $G$ is the sum of the component PSDs:

$$S_{GG}(f) = S_{SS}(f) + S_{NN}(f) = P_S(f) + P_N(f). \tag{14}$$

We analyze the CSD between the input $W$ and the output $G$. Using the distributive property of the CSD (derived from the linearity of expectation):

$$S_{WG}(f) = S_{W(S+N)}(f) = S_{WS}(f) + S_{WN}(f). \tag{15}$$

Since $W$ and $N$ are independent (and zero-mean), their CSD is zero ($S_{WN}(f) = 0$). Thus:

$$S_{WG}(f) = S_{WS}(f). \tag{16}$$

The magnitude squared coherency is defined as:

$$|C_{WG}(f)|^2 = \frac{|S_{WG}(f)|^2}{S_{WW}(f)S_{GG}(f)}. \tag{17}$$

We substitute the results from steps 2 and 3:

$$|C_{WG}(f)|^2 = \frac{|S_{WS}(f)|^2}{S_{WW}(f)(P_S(f) + P_N(f))}. \tag{18}$$

We now relate the numerator $|S_{WS}(f)|^2$ to $P_S(f)$. Recall that $S$ is the output of an LTI system with input $W$, without added noise. By Theorem 5.2 (Invariance of Coherency Magnitude under LTI Filtering), the magnitude squared coherency between $W$ and $S$ must be 1:

$$|C_{WS}(f)|^2 = \frac{|S_{WS}(f)|^2}{S_{WW}(f)S_{SS}(f)} = 1. \tag{19}$$

Therefore, we can express the numerator in Eq. 18 as:

$$|S_{WS}(f)|^2 = S_{WW}(f)S_{SS}(f) = S_{WW}(f)P_S(f). \tag{20}$$

Substituting this back into Eq. 18:

$$|C_{WG}(f)|^2 = \frac{S_{WW}(f)P_S(f)}{S_{WW}(f)(P_S(f) + P_N(f))}. \tag{21}$$

Assuming the watermark has power ($S_{WW}(f) > 0$), we simplify:

$$|C_{WG}(f)|^2 = \frac{P_S(f)}{P_S(f) + P_N(f)}. \tag{22}$$

The SINR is defined as $\text{SINR}(f) = \frac{P_S(f)}{P_N(f)}$. Dividing the numerator and denominator of the coherency expression by $P_N(f)$:

$$|C_{WG}(f)|^2 = \frac{P_S(f)/P_N(f)}{(P_S(f)/P_N(f)) + 1} = \frac{\text{SINR}(f)}{\text{SINR}(f) + 1}. \tag{23}$$

$\square$

