# OpenReview forum: "Remotely Detectable Robot Policy Watermarking"
_ICLR.cc/2026/Conference — ICLR 2026 Poster_

### Official Review · Reviewer_RB6n · 2025-10-26

**Soundness:** 4
**Presentation:** 4
**Contribution:** 3
**Rating:** 8
**Confidence:** 3

**Summary:**

This paper introduces CoNoCo (Colored Noise Coherency), the first framework for remotely detectable watermarking of robot control policies. Unlike prior approaches that require white-box or onboard access, CoNoCo embeds a frequency-domain signature into the stochastic exploration noise of a policy, enabling provenance verification from remote, noisy, and asynchronous observations (e.g., video footage). The watermark is injected by shaping the policy’s Gaussian noise into a narrow spectral band and later detected via spectral coherency, which is invariant to unknown system dynamics. The method is evaluated on simulated and real-world robots (e.g., RoboMaster platform) under various sensing modalities, showing high detectability, policy utility preservation, and strong robustness to additive noise attacks. The authors also open-source their full code and trained models.

**Strengths:**

The paper addresses a novel and important challenge: remote watermark detection for robotic policies, representing a new research direction beyond digital or model-based watermarking. The concept of bridging the “Physical Observation Gap” through spectral coherency is both elegant and well-motivated.

The methodology is clearly described and mathematically grounded, with supporting theorems demonstrating action-distribution preservation and LTI-invariant detectability. The experiments cover a diverse set of control environments and modalities, including real-world deployment, demonstrating consistent performance.

The paper is exceptionally well-written and organized. Figures clearly illustrate the concept and pipeline. The mathematical derivations are detailed yet readable, and limitations are candidly discussed in the appendix.

The problem addressed is highly relevant to both AI safety and IP protection in robotics. Remote verification of control policies is crucial for future large-scale autonomous deployments. The release of code and trained models enhances the paper’s impact and reproducibility.

**Weaknesses:**

While the paper convincingly demonstrates robustness under several noise conditions, it does not discuss or evaluate how the watermark performs under unseen or novel distortions beyond the tested scenarios. Real-world observation pipelines may involve motion blur, lighting changes, camera compression, occlusions, or domain shifts in dynamic, conditions under which the spectral coherency assumption might weaken. Without empirical evidence or analysis of these unseen distortions, it remains uncertain how broadly the robustness claim generalizes in practice.

**Questions:**

1. How does CoNoCo perform under unseen distortions such as camera compression, lighting variation, partial occlusion, or non-linear distortions that break LTI assumptions?

2. Could frequency-domain regularization or multi-band embedding improve robustness to these distortions?

3. Does the open-source release include pretrained detectors for other robot types or example video-based detection pipelines?

4. Would combining CoNoCo with learned feature-based detectors (e.g., neural coherence estimators) further enhance robustness under non-linear observation mappings?

---

> ### Author Response · Authors · 2025-11-20
>
> We sincerely thank the reviewer for their encouraging and constructive remarks and provide our responses below.
>
> **(Weaknesses and Question 1) Distortions.**
>
> Following this comment, as well as feedback from other reviewers, we have added new robustness experiments to test the limits of CoNoCo. These experiments are presented in the new Appendix G and evaluate CoNoCo’s detection abilities under the following increasing alterations:
> - **G1:** offset synchronization in the glimpses, where the auditor begins observing the system long after it starts operating.
> - **G2:** time jitter in the glimpses, where the time interval between two glimpses varies slightly.
> - **G3:** glimpse drop, where a proportion of the glimpses is removed.
> - **G4:** camera shift, where the camera or motion-capture setup acquiring the glimpses is shifted relative to the side or top view.
> We also include two new robustness analyses in Appendix H, which study additional types of attacks on our system, complementing the Gaussian noise attack already present in the paper.
>
> While these new experiments do not directly address the specific perturbations mentioned by the reviewer (motion blur, lighting changes, camera compression, occlusions), they provide a broad view of CoNoCo’s robustness. We note that our experiments rely on simple vision algorithms (e.g., template matching), and many of the perturbations highlighted by the reviewer could likely be addressed using more advanced vision methods. We are excited to explore this direction in future work.
> Finally, regarding non-linear distortions that break LTI assumptions, we highlight that our experimental setup already includes environments that are not LTI, such as HalfCheetah with contact forces, and our results demonstrate CoNoCo’s robustness in these scenarios. An extended discussion on this point is provided in Appendix C.
>
> **(Question 2) Frequency Domain Regularization / Multi-Band Embedding.**
>
> We appreciate this insightful suggestion. We agree that multi-band embedding would provide redundancy against band-specific noise (e.g., mechanical resonance), while frequency-domain regularization could improve robustness by suppressing frequencies with high signal-to-noise ratio. While our new experiments (Appendix G) demonstrate CoNoCo's existing robustness to significant time-domain and additive distortions, we view these mechanisms as excellent avenues to further extend the method's resilience to complex non-linear distortions in future work.
>
> **(Question 3) Open Source video analysis.**
>
> The open-source release includes all code used to produce our experiments, including the video-based detection pipelines for all results presented in the paper (this pipeline can be found in `watermarking_strategies/base.py’ file and in the ‘environments/’ folder). For navigation tasks, this code employs feature matching to track the robot’s body and infer its velocity, which is then used as glimpses. For actuated joint tasks, it tracks the two extremities of each joint to compute angular velocities, which are subsequently used as glimpses. In response to comments from other reviewers, we have added a new task of navigation with obstacles and released the corresponding video code, which uses the same pipelines as in the original experiments, demonstrating its generalizability.
>
> **(Question 4) Learned Feature-based Detectors.**
>
> We thank the reviewer for this insightful suggestion, which highlights a powerful direction for extending the CoNoCo framework to even more challenging sensing environments.
>
> The current implementation of CoNoCo utilizes analytical Spectral Coherency primarily due to its strong theoretical property of LTI-invariance (Thm 4.2). This provides a model-agnostic way to handle unknown system dynamics (C2), enabling detection across different robotic platforms without requiring system-specific training data.
>
> However, the reviewer correctly identifies that highly non-linear observation mappings (Gmap​ in Def. 2.1) present a distinct challenge. While we currently handle modalities like video by extracting hand-engineered features (e.g., velocity estimates), strong non-linearities can degrade the quality of these features. This weakens the detection signal, as classical coherency primarily captures linear relationships.
>
> This is precisely where "learned feature-based detectors" or "neural coherence estimators" could offer significant advantages. A promising hybrid approach would be to integrate a learned feature extractor upstream of the CoNoCo detector. This deep learning component could be trained to transform raw, complex glimpses into a latent representation where the relationship to the embedded watermark is effectively linearized. The existing spectral coherency mechanism could then operate robustly within this learned space. We view this as an exciting avenue for future work.

---

### Official Review · Reviewer_Mmbz · 2025-10-31

**Soundness:** 4
**Presentation:** 4
**Contribution:** 3
**Rating:** 8
**Confidence:** 3

**Summary:**

The paper presents an approach to encode IP protections into robot actions using colored noise coherency approaches. This adds noise to robot actions, preserving the marginal action distribution. The noise used is Colored Gaussian Noise (CGN), which replaces white noise used during exploration phase of the RL system training. Unlike dynamic watermarking methods for sensor inputs, this approach requires detectability and is not a strict defense against attackers - as such, it is also simple enough to be rapidly detectable. The paper proves that the CGN watermark detectability converges to unity as the 'glimpse sequence' of action data observed increases.

**Strengths:**

The paper is very theoretically sound, proving its claims in theory before moving to experimentation, in one case on real robot hardware. It is also thorough in its discussion on limitations, open questions and questions such as attack resilience of the CoNoCo approach.

**Weaknesses:**

For a reinforcement learning based paper that focuses on IP protections in robotics, it seems too thin on the experimental section to me, but otherwise is excellent.

**Questions:**

1. Could the authors comment on the complexity of robots such as multi-jointed arms that have to also conduct fine manipulation? Would CoNoCo be applicable there, given the smaller margin of error for those manipulations?

2. While the authors address the limitations of glimpse sequence sensor data quality requirements in the appendices, I would like to know if approaches involving signal reconstruction under noise would allow more robust watermarking without the need to fit the glimpse sequence length required to the periodicity of the signal (as mentioned in the periodicity part of open questions).

---

> ### Author Response · Authors · 2025-11-20
>
> We are grateful for the reviewer’s very encouraging comments and provide detailed responses to each of their points below.
>
> **(Weaknesses) Additional Environments.**
>
> In response to this comment and similar feedback from other reviewers, we have added results on an additional environment in Appendix I. Specifically, we include results for a variant of the Navigation task from VMas with 8 randomly placed obstacles. This navigation task further highlights that CoNoCo remains highly effective even in scenarios where the policy’s variance is reduced, such as near obstacles. We note that, following comments from other reviewers, we also added new experiments in Appendix G and Appendix H that study the robustness of CoNoCo to a wide range of glimpse alterations and adversarial attacks. These experiments further underscore the robustness and versatility of our method.
>
> **(Question 1) Applications with small variances.**
>
> In general, most watermarking strategies for stochastic policies become less reliable as entropy tends to 0. For example, similar situations occur in watermarking strategies in other domains, such as LLMs, where only a small set of tokens may be valid in certain cases. Consequently, we acknowledge that in some environments, the available actions may have limited variance in some specific states. To further evaluate CoNoCo in such scenarios, we added an environment in Appendix I.2 featuring obstacle avoidance. Even near obstacles, where the variance of feasible actions is reduced, CoNoCo maintains strong detection performance, demonstrating its robustness in low-variance conditions.
>
>
> **(Question 2) Signal Reconstruction Methods.**
>
> We thank the reviewer for this suggestion and see integrating signal reconstruction under noise as an exciting direction to further expand the applicability of CoNoCo. In response to feedback from other reviewers, we have added a new Appendix G studying multiple types of glimpse alterations, demonstrating that CoNoCo is already highly robust to many such changes. Despite already having high robustness, integrating signal reconstruction under noise may further strengthen our method, and we are interested in exploring it in future work.

---

### Official Review · Reviewer_g3WJ · 2025-11-01

**Soundness:** 3
**Presentation:** 3
**Contribution:** 3
**Rating:** 6
**Confidence:** 4

**Summary:**

This paper proposes Colored Noise Coherency, or CoNoCo, a watermarking strategy for robotic control policies that can be detected from remote observations. The core idea is to replace standard white exploration noise with band limited colored noise and to detect the resulting spectral signature through coherency in the frequency domain. The authors argue that coherency is invariant to unknown linear time invariant dynamics, which makes the detector robust to the physical observation gap between actions and sensed motion. The study reports strong detection from motion capture and monocular video while preserving the marginal action distribution and task reward.

**Strengths:**

- Clear problem formulation of remote watermark detection with only glimpse sequences and a careful breakdown of synchronization uncertainty, dynamics, and noise.
- A principled detector based on spectral coherency that is motivated by standard results in signal processing and that aligns well with the physical setting.
- Broad experimental sweep across simulated and real platforms with multiple sensing modalities, including top down and side view video, with anonymization tests and ROC based reporting.

**Weaknesses:**

##

- Watermarks are not detected in the presence of obstacles in the navigation task. It remains to see if the CoNoCo policy characteristics would be detectable in a general cluttered environment.
- Inability to Handle Time Offsets: This is a major operational weakness. The paper states that CoNoCo "does not handle large time offsets well" and that detection requires the "glimpse data recording needs to start near the beginning of the robot's operations". In any realistic scenario (like pulling CCTV footage), an auditor will be "tuning in" at an arbitrary time, not at the precise moment the robot was activated. Ideally the watermarking sequence $W_k$ would be ran in short intervals or a study of the time offset detection capabilities w.r.t. offset and detection length required would be included.
- The authors test a naive Additive Noise Attack in Appendix G. This attack involves adding White Gaussian Noise (WGN) to the policy's actions before execution but the attacker has no other objectives like maintaining performance.
    - An adversarial RL agent would not just add *random* noise. It could be trained with a multi-objective reward function:
        1. Maximize the original task reward (to maintain performance).
        2. Add random noise or change the policy in a structured way.
    This RL agent would learn to output a structured jamming signal that would most likely interfere with the spectral signature in the secret frequency band $\mathcal{B}$.
    - Another option is policy distillation [1,2] where the adversary learns to copy the behavior of the watermarked policy while maintaining performance. This could effectively change the policy and thus remove the watermark.

### References

[1] Policy Distillation, Andrei A. R. et al., 2015

[2] Refined Policy Distillation: From VLA Generalists to RL Experts, Tobias J. et al., IROS 2026

**Questions:**

1. In the robustness to adversarial additive noise experiments in Appendix G, is the adversary given any objective to maintain performance, for example a reward preserving constraint or penalty on deviation from nominal actions?
2. Does the detector ever raise false positives on non watermarked policies that naturally concentrate energy in the secret band due to task dynamics, and what priors or band selection rules mitigate this risk?
3. How sensitive is detection to modest drift in the policy execution rate during a single deployment, and can the search grid adapt online?
4. How is policy distillation or behavior cloning as a way to remove the watermarking detection capability?

---

> ### Author Response · Authors · 2025-11-20
> **Official Comment by Authors - Part 1**
>
> We thank the reviewer for their insightful feedback and answer all the points below.
>
> **(Weaknesses) Navigation with Obstacles.**
>
> Following the reviewer’s suggestion, we created a new Navigation environment with randomly placed obstacles and trained a policy for it. Our results show that CoNoCo achieves performance on this task comparable to the standard Navigation environments.
>
> **(Weaknesses) Time Offsets.**
>
> We are pleased to report that in Appendix G.1 of the revised paper, we address this limitation using a standard signal processing tool: the Generalized Cross-Correlation Phase Transform (GCC-PHAT). This simple addition to CoNoCo allows robust synchronization of observed glimpses with the watermark sequence, even when recording begins long after the robot starts operation. GCC-PHAT leverages the same LTI-invariance property as spectral coherency, estimating time delays via phase differences and making synchronization robust to unknown system dynamics (C2). Appendix G.1 presents the enhanced algorithm (Algorithm 2) and an empirical study (Figure 6) showing that while baseline CoNoCo detection degrades with increasing offsets, CoNoCo-Offset-Handling maintains near-perfect detection (AUC ≈ 1.0) even for large, arbitrary time offsets. This enhancement addresses the operational weakness identified in the original manuscript and suggests that offset handling is no longer a significant challenge for our method.
>
> **(Weaknesses and Question 4) Adversarial strategies.**
>
> The reviewer raises important questions about advanced adversarial attacks. While we acknowledge that a sophisticated adversary analyzing long-term spectral averages could potentially identify the secret band B, CoNoCo remains resilient even under this assumption. We have significantly expanded Appendix H to rigorously analyze the strategies an adversary might use if they identify the band, including two new numerical experiments:
>
> **(i) RL-based Structured Jamming:** The reviewer suggested an RL agent could learn a structured jamming signal $J_k$​ to interfere with the watermark $W_k$​. We argue rigorously in Appendix H.2 that this is infeasible: cancellation would require predicting the instantaneous phase of the pseudo-random sequence $W_k$, which is impossible without the secret key (seed). We provide a new visualization showing that the power spectral density of the combined signal is the sum of the individual signals, so the power always increases. This scenario reduces to the original Additive Noise Attack already analyzed in the manuscript in Appendix H.1.
>
> **(ii) Filtering (Band-stop Filter Attack):** Alternatively, the adversary could apply a band-stop filter to remove signals in B. However, such a filter removes all content in B: not only the watermark but also the original policy’s intended actions. This effectively becomes self-jamming and can degrade performance whenever those frequencies matter for the task. To study this trade-off, we added a new numerical experiment in Appendix H.1. We found that applying a strong filter lowers the detection score but leads to a substantial increase in policy distortion (37.5% increase in MSE between the filtered and intended actions). This shows that an adversary cannot remove the watermark via filtering without risking significant degradation of the policy’s performance.
>
> **(iii) Policy Distillation:** We agree with the reviewer that policy distillation could potentially remove the CoNoCo watermark by disrupting the pseudo-random sequence, and have added a new Appendix H4 discussing this adversarial approach. While distillation might be effective in some cases, it is not always feasible and requires specific conditions to be a practical adversarial strategy. In particular, it introduces two significant challenges:
>
> * IP-sensitive policies, such as foundation models and autonomous driving policies, are typically trained on large amounts of diverse and proprietary real-world data (often from multiple robots and task types) as well as synthetic simulator data. Collecting sufficient data to distill such policies from deployment on a single robot, a few robots, or even via an API (often subject to rate limits or significant costs) may be infeasible or prohibitively expensive.
>
> * Since distillation is inherently an imperfect approximation, it frequently leads to a performance drop, reducing the value of the stolen IP [1].

---

> ### Author Response · Authors · 2025-11-20
> **Official Comment by Authors - Part 2**
>
> **(Question 1) Gaussian Adversary objective.**
>
> In Appendix H.4, the adversary is implicitly constrained to not deviate too far from the nominal action, because the magnitude of the allowed deviation is controlled by the strength of the noise.  Figure 11 shows that noise levels sufficient to evade detection also destroy the policy's reward. As discussed above, targeted attacks (filtering/jamming) are also infeasible or highly detrimental to performance.
>
> **(Question 2) False positives on non-watermarked policies.**
>
> We wish to clarify that CoNoCo detects coherency (phase alignment) with the specific, secret pseudo-random sequence $W_k$​, not merely high energy in band B. Thus, it is highly unlikely that natural dynamics randomly align in phase with the owner's secret sequence and lead to false positives. In our experiments, our high ROC AUC values confirm a very low false positive rate across tasks.
>
> **(Question 3) Drift in policy execution rate.**
>
> Our current approach does not adapt the grid search online. However, to address the reviewer’s point on drift in policy execution rate, we added a new study in Appendix G.2 analyzing CoNoCo detection under time jitter. Here, jitter is defined as slight variations in the time interval between glimpses and is quantified as the standard deviation of these intervals. Our results show that CoNoCo remains robust to levels of jitter far exceeding those observed in our real-world system, thus highlighting its robustness to such distortions.
>
> **References.**
>
> [1] Cho, J. H., & Hariharan, B. (2019). On the efficacy of knowledge distillation. Proceedings of the IEEE/CVF international conference on computer vision.

---

### Official Review · Reviewer_UYqi · 2025-11-01

**Soundness:** 3
**Presentation:** 3
**Contribution:** 3
**Rating:** 6
**Confidence:** 3

**Summary:**

The paper aims to design watermark for trained robotic policies in the observation-only setting. That is, to verify the ownership of a robot’s policy using only remote observation such as videos or motion capture. To make this concrete, the authors identifies the “Physical Observation Gap,” which captures the three key challenges. The approach focuses on Gaussian policies.

The method CoNoCo adds a covert spectral signature into the exploration noise of the policy by concentrating energy in a secret frequency band. This modification preserves the marginal distribution over actions, ensuring task performance is unaffected. The detector uses it to reconstruct and compare the signature via spectral coherency, scanning over possible execution rates.

Theory shows the watermark remains statistically invisible per timestep but can be detected over time. Experiments across real and simulated environments including RoboMaster, VMAS, and MuJoCo tasks demonstrate strong detection performance, reward preservation, and some robustness to noise.

**Strengths:**

1. Well-scoped and original problem: The paper clearly frames a new challenge—verifying the ownership of a robot’s policy using only remote sensing (e.g., video), with no white-box access. The proposed “Physical Observation Gap” is realistic and well-formulated, addressing timing mismatches, unknown dynamics, and sensing limitations.

2. Simple but clever method: The idea to use colored Gaussian noise with energy concentrated in a secret frequency band is elegant. It avoids changing the marginal action distribution while enabling detectability through spectral analysis. The implementation is straightforward and practical.

3. Solid theoretical grounding: The paper gives intuitive and mathematically sound analysis. It shows that marginal action distributions are preserved and that the coherence metric used for detection has a direct relationship with SINR.

4. Strong and diverse experiments: The authors validate the method in both simulation and a real robot setting, using various sensing modalities including motion capture and single-camera video. They report strong detection results, reasonable robustness, and include anonymity comparisons with a baseline.

**Weaknesses:**

1. Limited attack robustness: The experiments mainly test additive noise. But real-world attackers might apply frame drops, time shifts; none of which are evaluated here. These could undermine coherency-based detection.

2. Scope is restricted to continuous Gaussian policies: There’s no discussion on how this approach might extend to discrete or deterministic policies, which are common in practice

**Questions:**

1. Attack resilience: How does your method perform under time distortions, missing frames, or camera shifts? Any strategies to make it more invariant?

2. Beyond Gaussian policies: Do you see a way to adapt this idea to deterministic or discrete-action policies while retaining reward preservation and detectability?

---

> ### Author Response · Authors · 2025-11-20
>
> We thank the reviewer for their insightful and encouraging comments and provide detailed responses to each point below.
>
> **(1) Attack Resilience.**
>
> We have incorporated in a new Appendix G an empirical study of the alterations suggested by the reviewer:
> - **Time shifts / Time distortion:** Appendix G.1 examines the impact of glimpse offsets, and Appendix G.2 analyzes the effect of time-jitter in glimpses. Our results show that CoNoCo remains robust even under levels of time jitter far higher than those observed in our real-world system. We also introduce a simple mechanism that improves robustness to glimpse offsets and show that it achieves strong performances.
> - **Frame drops/Missing frames:** Appendix G.3 examines the detection score of CoNoCo under varying proportions of frame drops and shows that it remains robust to a high proportion of drops.
> - **Camera shifts:** Appendix G.4 simulates camera shifts using random planar projections of real-world data collected on our real-world RoboMaster platform. The results demonstrate that our frequency-based detection approach remains robust across a wide range of such distortions.
>
> Additionally, following the suggestions from other reviewers, we have extended Appendix H on Adversarial Robustness Analysis to include two more adversarial attacks. Namely, we have included filtering attacks (H.2) and structured jamming attacks (H.3). Our results in these settings further underscore the robustness of CoNoCo.
>
>
> **(2) Beyond Gaussian Policies.**
>
> We acknowledge that our proposed CoNoCo method is limited to scenarios where policies output a distribution over actions and does not handle deterministic policies. At the same time, we emphasize that stochastic policies are common and often essential in continuous robotic control domains. Recent foundation models for robotics—which would be natural targets for watermarking with CoNoCo—have employed stochastic policies [1, 2]. Following the reviewer’s comment, we have explicitly added this point to the detailed limitations in Appendix A.
>
> To address the reviewer’s question on how one could achieve this, one potential direction is to define distributions at a higher level of abstraction and sample from behaviors rather than individual actions—for example, watermarking entire trajectories instead of single control signals in robotic navigation.
>
>
> **References.**
>
> [1] Black, K., Brown, N., Darpinian, J., Dhabalia, K., Driess, D., ... & Zhilinsky, U. π0. 5: a vision-language-action model with open-world generalization, 2025. URL https://arxiv. org/abs/2504.16054.
>
> [2] Liu, M., Pathak, D., & Agarwal, A. (2025). LocoFormer: Generalist Locomotion via Long-context Adaptation. Conference on Robot Learning.

---

### Author Response · Authors · 2025-11-20

We thank the reviewers for their thoughtful comments. We are encouraged that the reviewers found our problem 'well-scoped and original' (Reviewer UYqi), our method 'elegant and well-motivated' (Reviewer RB6n), and our theoretical and experimental contributions 'very theoretically sound' (Reviewer Mmbz), 'mathematically grounded' (Reviewer RB6n), and supported by a 'broad experimental sweep' (Reviewer g3WJ). In response to their feedback, we have made several updates to the revised manuscript to further strengthen our results:
- **Clarified open questions:** We addressed limitations regarding deterministic policies and real-world sensing constraints (Appendix A).
- **Additional experiments on robustness to alterations:** We conducted a comprehensive empirical study on glimpse alterations (time offsets, jitter, frame drops, camera shifts) in Appendix G.
- **Time offset handling:** We show that a minimal transformation (GCC-PHAT) to the glimpse data before it is inputted to CoNoCo enables it to robustly handle large time offsets, retaining perfect AUC (Appendix G). Hence, this is no longer a limitation of our method, and we revised the paper text accordingly.
- **Expanded analysis of adversarial attacks:** We analyzed two additional adversarial attack scenarios, Band-Stop Filtering and Structured Jamming, demonstrating the trade-off between attack success and policy utility in Appendix H. We also consider distillation in the same appendix.
- **New environment:** We validated our method on a new environment: VMAS Navigation with obstacles, confirming performance in constrained settings (Appendix I).

We are grateful to the reviewers for the many constructive insights, which have significantly strengthened our manuscript, and we welcome any further questions or suggestions during the discussion phase.

---

### Meta-Review · Area_Chair_Uhtv · 2026-01-02

**Summary:**

I suggest acceptance for this paper, informed by the following reviewer feedback. Reviewers **UYqi**, **g3WJ** and **RB6n** state that the *strengths* are a clear problem formulation of remote watermark detection with many relevant empirical results, addressing a novel and timely challenge.  Reviewers **UYqi** and **g3WJ** also comment on the theoretical soundness of the work.

**UYqi** states the main *concerns* are (1) limited attack robustness and (2) limited to continuous Gaussian policies.

**g3WJ** states that the main *concerns* are (1) unclear if applicable to cluttered environments, (2) need to handle time offsets and more adversarial attacks.

**Mmbz** states that the main *concerns* are that for an RL paper, it appears “too thin on the experimental section”.

**RB6n** states that the main *concerns* are that it does not discuss or evaluate how the watermark performs under unseen or novel distortions beyond the tested scenarios.

The authors have provided many more experiments and discussions during the rebuttal to address these concerns.

**Reviewer Concerns:**

**UYqi Concerns**
* (1) limited attack robustness $\rightarrow$ **addressed (empirically)**. Added new results in Appendix G with two more adversarial attacks and three others suggested by the reviewer.
* (2) limited to continuous Gaussian policies $\rightarrow$ **addressed (discussion)**.


**g3WJ Concerns**
* (1) unclear if applicable to cluttered environments $\rightarrow$ **addressed (empirically, added new environment with clutters)**
* (2) need to handle time offsets and more adversarial attacks $\rightarrow$  **addressed (empirically, added method to handle time offsets and tested more adversarial attacks)**

**Mmbz Concerns**
* (1) Thin experimental section $\rightarrow$ **addressed (empirically, added results on an additional environment in Appendix I)**

**RB6n Concerns**
* (1) performance in unseen or novel distortions $\rightarrow$ **not addressed**. The authors did add new results in Appendix G with two more adversarial attacks and three others suggested by the reviewer. However, the authors acknowledged “While these new experiments do not directly address the specific perturbations mentioned by the reviewer (motion blur, lighting changes, camera compression, occlusions), they provide a broad view of CoNoCo’s robustness.”

**Reviewer Scores:**

* **UYqi** would have *maintained* a score of 6: marginally above the acceptance threshold.
* **g3WJ** would have *maintained* a score of 6: marginally above the acceptance threshold.
* **Mmbz** would have *maintained* a score of 8: accept, good paper (poster)
* **RB6n** would have *maintained* a score of 8: accept, good paper (poster)

---

### Decision · Program_Chairs · 2026-01-26

Accept (Poster)